# Blockchain of Carbon Trading for UN Sustainable Development Goals

**Seong-Kyu Kim [1] and Jun-Ho Huh [2,*]**

[1]   Department of Information Technology, Sungkyunkwan University, Seoul 03063, Korea; guitara7@skku.edu
[2]   Department of Data Informatics, Korea Maritime and Ocean University, Busan 49112, Korea
*    Correspondence: 72networks@kmou.ac.kr

**Abstract:** Carbon credits should reduce the environmental pollution and carbon emission of the Earth in the future. The market for carbon credits will become a critical issue from 2021, and carbon credits will be applied to systems where individuals can trade. In order for these carbon credits to be traded between individuals, however, a corresponding exchange of carbon credits is needed. Policies, strategies, and technologies are also necessary to measure the trading of carbon credits. This paper aims at making transactions more reliable by applying blockchain technology to measure carbon emission rights. It uses blockchain to verify carbon emissions rights among the UN-SDGs' (United Nations Sustainable Development Goals') 17 tasks. In addition, it introduces the necessary dApp. In fact, we can protect against carbon emissions anomalies by using big data and artificial intelligence in mobile cloud environments. Thus, this paper proposes a blockchain-based carbon emission rights verification system to learn proven data further by using the governance system analysis and blockchain mainnet engine to solve these problems.

**Keywords:** blockchain; artificial intelligence; information security; UN sustainable development goals; UN SDGs; carbon trading; dApp

---

## 1. Introduction

The Kyoto Protocol is a market-based mechanism that formed the framework of the international climate change response system, as described by the United Nations, to ease the burden of cost of greenhouse gas reduction activities by the mandatory reduction bureau. The Kyoto Method consists of carbon emissions trading (ET), a clean development mechanism (CDM), and a joint implementation system (JI); in particular, carbon emission trading (emissions trading) refers to the market wherein greenhouse gas emission rights (emissions trading) are acquired. Where carbon emission rights cover allocations and credits 1), quotas refer to greenhouse gas emission rights paid to major greenhouse gas sources, such as power generation facilities or production facilities, and credits pertain to unit-to-consumption business projects projected for external greenhouse gas reduction projects (BAU-U, BAS-US, etc.). Meanwhile, "market" means that the price of carbon credits is determined by the demand and supply of carbon credits in the market instead of being fixed by policy. This is done in a way that reflects the costs of producing goods or services as the environmental and social costs of climate change [1].

This is in contrast to the carbon tax, wherein the size is determined. According to the World Bank's tally, the carbon emission trading market reached about 10.9 billion USD in 2005 when the Kyoto Protocol took effect and continued to grow at an annual rate of 108%, growing to 143.7 billion USD in 2009. Meanwhile, the growth of the carbon emission trading market slowed down a bit in 2009 when economic activity shrank in the wake of the global financial crisis. Based on the recovery outlook for the global economy, however, the global carbon emission trading market is expected to

reach about 669 billion USD in 2013, according to a report. The carbon emission trading market can be classified in various ways. First, depending on the nature of the carbon emission rights, it can be largely classified as a quota market and a credit market. As shown in Figure 1, Retail allows users to buy and sell goods, energy, etc. using Carbon Impact. In addition, Back-End signed an electronic contract for Carbon Credit, which is a smart contract that is generated by UserProfile, and can use trading rights for carbon transactions in all mobile apps, called Ecosystem. Credit markets are often referred to as project-based markets, which, in turn, can be divided into primary markets and secondary markets. On the other hand, depending on the way the transaction is made, the carbon emission trading market can be classified into OTC (Over-the-Counter) markets and in-house markets. These carbon credits translate into more and more transactions, and they are divided into 17 important subdivisions in the future SDGs (Sustainable Development Goals) at the United Nations. To recognize these carbon credits, carbon credits are accurately measured using non-modifiable blockchain technology and, by extension, proposed by the UN's (United Nations) SDGs as the environmental factors referred to by the UN's SDGs [2].

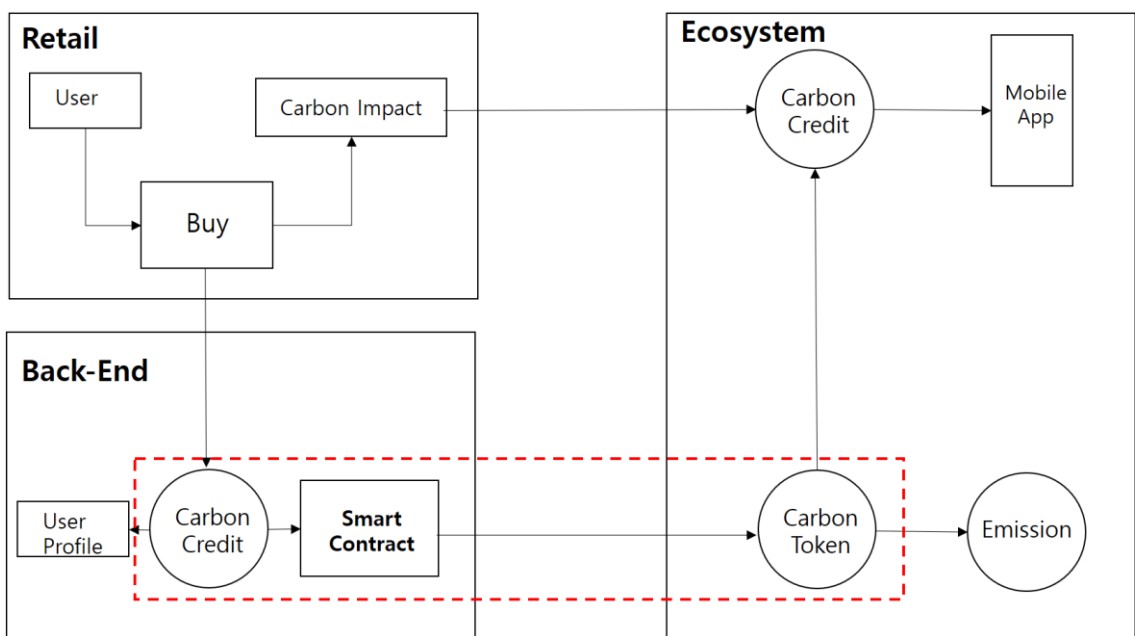

**Figure 1.** Smart renewable energy and P2P (Peer-to-Peer) blockchain service.

The reason is that the governance system is applied to the carbon credits based on the accuracy of blockchain according to its stability, decentralization, and reliability. To overcome these problems, a hybrid blockchain is proposed. In addition, we use blockchain to verify carbon emission rights in our efforts to manage energy [3]. These efforts are taken a little further in the verification system for blockchain. Therefore, in order to prevent hacking and strengthen security vulnerabilities, the available blockchain network protocol will be applied and changed to the appropriate blockchain network protocol for the greenhouse gas comprehensive information center. In addition, this study provides a scope for the purpose of verifying carbon emission rights using blockchain to conduct transactions between individuals and for the performance of trading of carbon emission rights. The method also aims to instantiate blockchain performance data, Transactions per Second (TPS), and to ensure the transparency and integrity of carbon credits for future P2P (Peer-to-Peer) transactions.

Section 1 discusses the problems of carbon emissions in the UN-SDGs and the artificial intelligence blockchain for verifying the trading rights when future carbon emissions are traded between individuals. When we introduce this blockchain, we talk about the important consensus algorithm. Section 2 presents directions on how to apply blockchain in the UN-SDGs from realized work, and specifically

discusses Information Technology (IT) application and blockchain application for carbon credits and renewable energy. Section 3 describes the UN SDGs' Performance and Blockchain Algorithms for Design and Implementation. Section 4 shows configurations for performance comparison. The actual performance comparison shows 15,000 TPS. Section 5 discusses the direction of future research and current problems. Section 6 concludes with a conclusion on the study of the application of blockchain carbon credits.

## 2. Background Knowledge

Studies related to blockchains and energy trading have been conducted by several researchers. M. Andoni et al. published their research work titled "Blockchain Technology in the Energy Sector: A Systematic Review of Challenges and Opportunities," whereas S. Wang et al. came up with "Energy Crowdsourcing and Peer-to-Peer Energy Trading in Blockchain-Enabled Smart Grids." F. Luo et al. and K. Gai et al. also released "A Distributed Electricity Trading System in Active Distribution Networks Based on Multi-Agent Coalition and Blockchain" and "Privacy-Preserving Energy Trading Using Consortium Blockchain in Smart Grid," respectively [4–6].

### 2.1. Carbon Emission

A certified emission reduction (CER; certification reduction or public certification reduction) refers to a confirmation made by the UN that a Clean Development System (CDM) project reduced greenhouse gas emissions. These carbon credits can be traded on the market through the emission trading system. If advanced countries go to developing countries for greenhouse gas reduction projects, the UN will examine and assess them and give them a certain amount of carbon emission rights. This greenhouse gas reduction project is called the Clean Development System (CDM) project. Not only advanced countries, but also developing countries themselves can carry out CDM projects to acquire carbon emission rights, which the Republic of Korea does. This refers to companies in each country that have failed to reduce carbon dioxide (CO2) emissions within a set period of time, paying money to buy rights from businesses that have been able to afford emissions or grow forests. Under the Kyoto Protocol, parties were required to reduce their carbon dioxide emissions by an average of 132% from 2008 to 2012, based on the 1990 emissions [7]. Countries that have succeeded in reducing their emissions have been allowed to buy and sell the corresponding carbon credits. In other words, petrochemical companies and other companies emitting a lot of carbon dioxide must reduce their carbon dioxide emissions themselves or buy rights from forest land owners in countries with low emissions. The Republic of Korea introduced such carbon emission rights from 2015 to 2018 (see Figure 2).

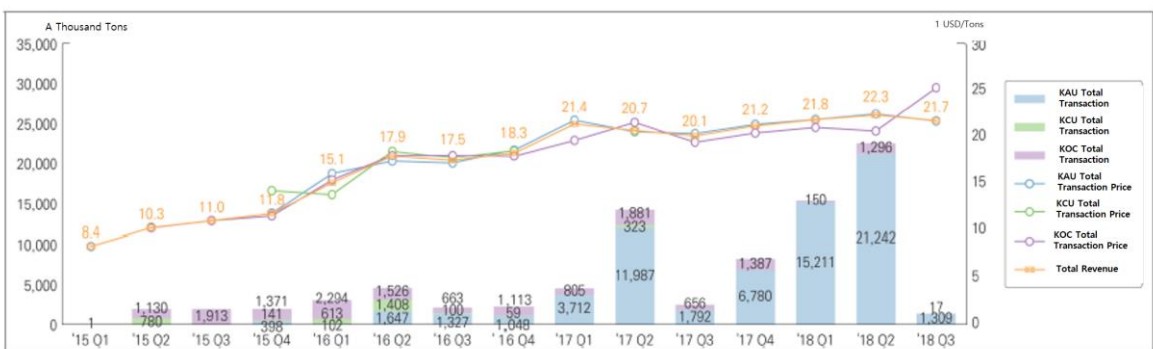

**Figure 2.** Growth rate of carbon emissions [8].

In addition, the total volume of emissions traded in both in-house and over-the-counter transactions during the branch line result period (2015.1–2018.8) was 86.2 million tons, with 37.5 million tons and 48.7 million tons traded in disability and over-the-counter transactions, accounting for 44% and 56%,

respectively. By emission rights, 66.6 million tons of Korean Allowance Units (KAU), 3.4 million tons of Korean Credit Unit (KCU), and 16.2 million tons of Korean Offset Credit (KOC) were traded, representing 77%, 4%, and 19% of the total. The total volume of transactions per year increased by 208%, 246%, and 134% year-on-year to 5.7 million tons in 2015, 11.9 million tons in 2016, 29.3 million tons in 2017, and 39.2 million tons in 2018, and the last year was similar to 2018, reflecting only the performance of transactions in the first half of the year. In addition, during the same period, the average price increased by 156%, 122%, and 106% year-on-year to 17,1799 KRW in 2016, 20,897 KRW in 2017, and 22,127 KRW in 2018, ending with a double increase compared to the initial average price in 2015. The average transaction price for the entire trading period was 20,279 KRW. In addition, by emission rights, KAU was traded at 15,767 KRW, KOC was 16,703 KRW, and KAU was traded at a relatively higher price than other emission rights, and the market price was slightly higher than the over-the-counter price. Continued increases in transaction prices and expansion of trading volume also affected transaction payments, rising 324%, 333%, and 142% year-on-year to 204.4 billion KRW in 2016, 612.3 billion KRW in 2017, and 868 billion KRW in 2018 from 63.1 trillion KRW in 2015; combined, the total transaction amount was 1.7477 trillion KRW. KAU accounted for 81%, 3%, and 15%, respectively, with 1.4231 trillion KRW; KCU accounted for 54 billion KRW and KOC for 270.6 billion KRW, while the total transaction amount in the market and over-the-counter trading market was 781 billion KRW and 966.7 billion KRW, respectively, representing 45% and 55% of the share, similar to the share of the trading volume [8].

Carbon emission credits refer to the right to emit greenhouse gases, one of the substances giving the Earth's environment a load. When humans emit carbon dioxide ($CO_2$), nature absorbs it and keeps the $CO_2$ concentration in the atmosphere at a certain level; when the emission exceeds Earth's own capacity to conserve the environment, however, $CO_2$ concentration increases gradually, resulting in global warming and climate change. Global $CO_2$ emissions rose 1.4% (4.6 billion tons) year-on-year to 32.5 billion tons in 2017 from 32.5 billion tons (see Figure 3) [9]. Kyoto Protocol 2, which took effect in 2005 by giving companies the right to emit greenhouse gases only at a manageable level, is set at 41.8 billion tons, or about 5.2% less greenhouse gas emissions from developed countries. The Emission Trading Scheme is a system that uses the advantages of the market mechanism to reduce greenhouse gases to maximize social costs, and the emission trading system is expanding worldwide, including in Korea, allowing companies to trade their remaining or insufficient emission allowances freely [10,11].

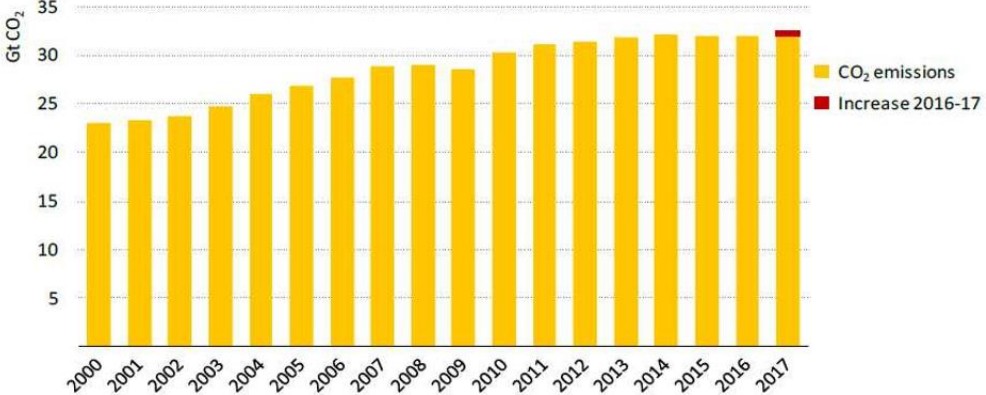

**Figure 3.** Trend of carbon dioxide ($CO_2$) emissions related to global energy. Source: Korean Government; "Second National General Management Plan for Greenhouse Gas Statistics (2020–2024)"; KEA Brief Issues of Energy, 2018 [8].

Blockchain technology is considered to be a suitable technology for implementing an emission trading platform that enhances the efficiency, security, and transparency of transactions, since it enables the establishment of multiple trust networks among stakeholders without the need for a Trusted Third Party.

In particular, the effective linkage of emission markets operating in different countries/regions requires the use of blockchain technologies that implement global networks involving multiple members without the need for centralized management. In addition, the climate change response system is changing from the top-down method of allocating greenhouse gas emission rights to developed countries only to the bottom-up formula of self-determination of national reduction targets and methods involving both developed and developing countries. This is a common task of reducing greenhouse gas emissions by establishing a blockchain-based platform wherein the majority of countries around the world participate as members to trade emission rights on their own and jointly verify, record, and store them.

Using this decentralized blockchain technology, the information and transmission details of carbon emission rights can be utilized to incorporate the technologies of blockchain and have safer transactions. In the future, carbon credits are expected to be traded using personal communication in P2P. In the EU (European Union), individuals buy and sell carbon credits. To ensure the safe trade of these carbon credits, we suggest a methodology applied with blockchain technology [12,13].

## 2.2. UN Sustainable Development Goals

The Sustainable Development Goals (SDGs), as the agenda that the 70th UN General Assembly held in 2015 resolved to achieve by 2030, are 17 shared goals for realizing the ideology of sustainable development. The 2030 Sustainable Development Goals (SDGs), together with the slogan "Leave no one behind," consist of 17 goals and 169 detailed goals for humanity in five areas: Human, earth, prosperity, peace, and partnership [14,15]. In 2015, the expiration of the MDGs' (Millennium Development Goals) implementation target deadline required governments to continue their efforts to achieve their goals and address new issues. Over the past 15 years, the United Nations has been discussing what the global priorities should be. At the Rio+20 meeting in June 2012, we agreed on the post-2015 global development system and came up with 17 new goals or global priorities: Sustainable Development Goals (SDGs) [16].

The MDGs were applied to all countries in principle; in practice, however, the goal was focused on developing countries. SDGs call for all countries, including developed, developing, and underdeveloped, to work toward the prosperity of humankind and to protect the environment. The MDGs were a useful development agenda but had too narrow of a target range, whereas the SDGs are even more comprehensive. The Sustainable Development Goals (SDGs) are flexible in a variety of national situations, allowing countries to select and measure detailed goals and indicators within their most relevant objectives. Even now, countries around the world are making great efforts to implement the Sustainable Development Goals (SDGs). Korea in particular is implementing individual UN-SDGs through government policies and related laws, such as the Framework Act on Sustainable Development, Framework Act on Low-Carbon Green Growth, and Framework Act on International Development and Cooperation [17].

### 2.2.1. Clean Energy

Energy is the core of the major challenges and opportunities facing the world today. Energy use is essential for jobs, security, climate change, food production, or income growth. Trying to achieve this goal is particularly important because it is linked to other sustainability goals. Focusing on the generalization and efficiency of energy, the increasing use of renewable energy through new economies and job creation will help create sustainable and inclusive communities and address environmental issues such as climate change. Currently, three billion people are at risk of air pollution due to the inability to use clean cooking utensils. Likewise, less than a billion people live without electricity, and 50% of them are found in sub-Saharan Africa. Fortunately, over the past decade, there has been great progress in the use of hydroelectric, solar, and wind power, and energy use per unit of GDP has decreased. Nonetheless, the challenges at hand have not yet been resolved. We need greater access to clean fuel and technology, and more progress needs to be made to integrate renewable energy into the

final energy of buildings, transportation, and industry. Public and private investments in energy need to be increased, with focus on regulatory regimes and innovative business models to change the global energy system [18].

### 2.2.2. Response to Climate Change

Climate change is now affecting all countries across the continent. It is disrupting the national economy, affecting life, and causing great losses to humanity, communities, and countries today and in the future. Weather patterns are changing, sea levels are rising, climate change is frequent, and greenhouse gas emissions are now at their highest levels. If no action is taken, sea level temperatures are expected to rise above an average of three degrees this century. This has the most effect on the poor and vulnerable. By seeking economic and scalable solutions, the nation can become a cleaner and more resilient economy. The pace of change is accelerating as more people seek ways to use renewable energy and reduce and adapt to greenhouse gas emissions. However, climate change is a transnational global challenge. We need an international level of improvement to help developing countries move toward a low-carbon economy. To strengthen the international response to the threat of climate change, countries adopted the Paris Convention at the Conference of Parties to the United Nations Framework Convention on Climate Change (COP 21), which took effect in November 2016. Under the agreement, all countries agreed to limit the Earth's temperature rise to less than two degrees Celsius. In April 2018, 175 countries ratified the Paris Agreement, and 10 developing countries submitted their first national response plans to combat climate change.

### 2.2.3. Poverty Eradication

The global poverty rate has been halved since 2000, but one in 10 developing countries still lives below the 1.9 USD-a-day international poverty line, and there are millions of people making less than a day's living expenses. Despite outstanding progress in East Asia and Southeast Asia, 42% of sub-Saharan Africa's population is still suffering from extreme poverty. Poverty means more than just a lack of income and resources to ensure a sustainable livelihood. These include hunger and malnutrition, restrictions on education and living services, social discrimination and exclusion, and restrictions on decision-making participation. Economic development must have a comprehensive goal to provide sustainable jobs and improve the structures of inequality. Social protection needs support to alleviate the suffering of countries at risk of disaster and to overcome economic crises, and these systems will help strengthen the responsiveness of those suffering from unexpected cost losses in the event of disaster and, ultimately, end extreme poverty in areas of absolute poverty.

### 2.2.4. Famine Species

It is time to rethink how we grow, distribute, and consume food. If done properly, agriculture, forestry, and fishing can provide everyone with nutritious food, generate a significant level of income, and, at the same time, support people-centered rural development and protect the environment. Now, biodiversity is rapidly decreasing due to damage to soil, fresh water, sea, and forests. Climate change is having a devastating effect on the resources we depend on, and it is increasing the risk of disasters, such as drought and flooding. Many farmers are no longer able to make ends meet on their land and have to move to cities to find opportunities, and poor food security causes serious undernourishment, hurting the growth of millions of children or shortening their lifespan. The world's undernourished population is expected to be 815 million, and an additional 2 billion by 2050, which requires fundamental changes in the world's food and agriculture systems. Investment in agriculture is essential to the development of agricultural productivity, and sustainable food production systems are necessary to reduce the risk of malnutrition.

### 2.2.5. Health and Well-Being

Ensuring a healthy life for all ages and promoting welfare are essential to sustainable development. While significant progress has been made, including increasing human life expectancy and decreasing infant and pregnant women's death rates, a professional improvement in the delivery system is needed to reduce the death rate of fewer than 70 children per 100,000 people by 2030. In order to achieve the goal of reducing early death rates from non-inflammatory diseases by one-third by 2030, efficient techniques are encouraged, such as the use of clean oils in cooking, and training on the dangers of cigarettes is needed. It also takes a lot of effort to eradicate a variety of diseases and to address the ongoing health problems. By focusing on providing effective funding for healthcare systems, improving sanitation, increasing access to medical services, and providing information on preventing environmental pollution, we can make great strides in saving millions of lives.

### 2.2.6. Quality Education

Quality education is the foundation for sustainable development. In addition to improving the quality of life, providing comprehensive education also helps local people have the creative thinking they need to find innovative solutions to international issues. Currently, more than 265 million children have dropped out of school, and 22% of them are only elementary school students. In addition, even children who go to school lack basic skills, such as reading and calculating. Over the past decade, efforts have been made to increase access to full-course education and to increase women's school enrollment. The eradication of basic illiteracy has increased dramatically, but more effort is needed to achieve universal educational goals and progress. For example, while the equality of primary education between men and women has been achieved, few countries have achieved that goal in the whole course of education. The reason for the lack of quality education is linked to capital issues, such as the lack of trained teachers, poor school facilities, and relatively poor opportunities for rural children. In order to provide quality education to children from poor families, we need to invest in educational scholarships, teacher training workshops, school building and water quality improvement, and school electrical facilities.

### 2.2.7. Gender Equality

While the Millennium Development Goals (including achieving universal primary education, MDGs) have made progress in gender equality and women's rights and interests, women and girls continue to suffer discrimination and violence throughout the world. Gender equality is not only a basic human right, but a necessary foundation for a sustainable world that pursues peace and prosperity. One in five women between the ages of 15 and 49 stated that they had been victimized by an intimate partner within 12 months of the survey date, and 49 countries currently do not have laws to protect women from domestic violence. Over the past decade, child marriages have been on the decline and progress has been made, with the harmful practice of female genital resection (female circumcision) decreasing by 30%, but more intensive efforts are needed to eradicate this practice. Providing women and girls with education, health care, quality jobs, and increasing their participation in the political and economic decision-making process will bring overall benefits to the sustainable economy, society, and humanity. Therefore, establishing a new legal system for equality of women in the workplace and eradicating harmful practices against women is critical to ending gender discrimination in many countries around the world.

### 2.3. Blockchain

Blockchain is a distributed computing technology-based data tampering prevention technology. This is a technology that stores managed data in a distributed data storage environment called a "block", where small data are linked in chain form based on the P2P method, so that no one can arbitrarily modify it and anyone can access the results of the change. The block also records all transactions that

were propagated to users prior to the discovery of the block, which is sent equally to all users in a P2P manner, so that the transaction details cannot be modified or omitted arbitrarily. Blocks have links to the date they were found and previous blocks; a collection of these blocks is called a blockchain [18–21].

Simply put, it is a technique that merely sets a bunch of records together. Unlike keeping transaction records on a central server when trading e-money in the past, blockchain shows the transaction records to all users and compares them to prevent forgery. As shown by Bitcoin's initial demonstration of the concept of blockchain and Ethereum's initial implementation of the concept of smart transport, there is a close relationship between blockchain and cryptocurrency. Still, blockchain is not a technology that can be used for cryptocurrency only. Cryptocurrency is said to be subordinate to blockchain.

Thus, technologies or services that have already applied blockchain are being developed. Interestingly, Satoshi Nakamoto solved the problem of developing and applying blockchain instead of developing blockchain first and applying it to Bitcoin, an electronic money system operated only by P2P. This can be seen in "Bitcoin: A Peer-to-Peer Electronic Cash System," the white paper released at the time of the unveiling of Bitcoin to the world. Largely divided into public blockchain, Private blockchain, and concourse blockchain, Bitcoin, which can be seen as the first implementation of the first public blockchain, introduced a proof-of-work approach to verify that the transaction history contained in each "block" was not tampered with by a malicious attacker and to leave only normal blocks. This is done in "proof" fashion, demonstrating that one of the many nodes (partners of the blockchain network) has succeeded in creating a new block that is directly connected to the old blocks (in a chain), which correspond to a work. All nodes (users) participating in the Bitcoin network have the right to create blocks to follow. Still, the next block, considered part of the normal chain on the network, will result in only one. Bitcoin mining refers to the process of changing the non-node value of the block header of new "block candidates" until the result of the execution is less than the "difficulty" value set in the Bitcoin network. The new block candidate will then contain the transaction details made since the last block creation [22–24]. Under this architecture, if more than 50% of nodes are "honest nodes"—a healthy node that does not continue to span blocks containing erroneous transactions—Bitcoin will maintain integrity. If more than 50% of the computing power consumed in mining is an "attacker node" that spans blocks containing erroneous transactions, the integrity of Bitcoin can be destroyed, often referred to as a "51% attack".

The second one is a private blockchain, which refers to a blockchain network where only authorized users (nodes) can participate. If the entire trading history on the network is made transparent, it can be introduced around troubled financial sectors and others [25]. The key feature is that it does not rely on cryptocurrency and, ironically, the reason it exists is that nodes that have validated the validity of a blockchain need a medium for payment. In a private blockchain, an unspecified number of people need not adopt an encryption-waste system because it does not have to serve as a node. There is also a point where the processing speed is fast against the public block chain. This is due to other variance consensus algorithms. Typical solutions include the Linux Foundation, IBM's Hyperledger Fabric, and the Hyperledger Iroha managed by the Linux Foundation and under development by a Japanese company [26–30], Corda of R3. Hyperledger is a blockchain project managed by the Linux Foundation, jointly developed by IBM and Digital Asset, and released as open-source [31]. It enables implementation with the actual code, and IBM provides blockchain solutions based on this project. In the case of a private blockchain by a single entity, it has been regarded as a slow database system with no advantage over a server-based database. Blockchain's security algorithms are aimed at keeping the same data among unreliable players because, if there is one entity, it is only a waste of computation and time. In defense against external attacks, the same effect can be achieved by using a server-based distributed system [32–36], and the scalability, convenience, economy, and performance are also superior to those of the server. Finally, there is the Consortium Blockchain, which combines public and private blockchains [37–45] (see Table 1).

**Table 1.** Types of blockchains.

|  | Public Blockchain | Private Blockchain | Consortium Blockchain |
|---|---|---|---|
| Management Subject | All Participants | Managed by the Central Institution | Participants in the Consortium |
| Network Participating Condition | None | Managed by the Central Institution | None or managed by a selected institution |
| Transaction Speed | Slow | Fast | Fast |
| Identification | Anonymous | Identifiable | Identifiable |
| Transaction Proof | Proof of work algorithm, Transaction verifier cannot be known in advance | Transaction verification is made by the central institution | Transaction verifier is known through certification, Transaction verification and block |

Source: Software Policy Institute, 2018 [25].

The rapid increase in world energy demand over the last decade and the requests for sustainable development can be approached through micro- and nanogrids using hybrid power systems based on the energy internet, blockchain technology, and smart contracts. In this way, renewable energy sources, fuel cell systems, and other energy generating sources will be optimally combined and connected to the grid system using advanced energy transaction methods [46]. In particular, this paper's contribution has the point that carbon credits should reduce the environmental pollution and carbon emissions of the Earth in the future. While the size of the carbon emission trading system has grown to a low point, there is a lack of technology to limit it, so research has begun. So, we are talking about the blockchain process for reducing greenhouse gas (GHG) and the measurement, reporting, and verification (MRV) methodology, as well as the verification process in a total of five stages.

For Protocol Design for Hybrid Governance, blockchain talks about how to use and verify IT governance. A procedure for verifying and agreeing on a carbon-tracking AI blockchain platform and a methodology for applying a hybrid governance protocol applicable to the UN SDGs are presented. The procedures of certificate-based hybrid blockchain and authentication of hybrid blockchain based on a cryptographic protocol are applied by applying this methodology: Certification of hybrid blockchain based on MAC (Media Access Control Address) Address, Authentication of Hybrid Blockchain based on ID/password, Hybrid Governance Protocol for Design Pattern, Hybrid Governance Protocol for Visitor Pattern, Hybrid Governance Protocol for Governance Protocol for Protocols, and Hybrid Governance Protocol for Protocols. It will then unveil the source code for the implementation of the hybrid governance protocol.

Thus, this paper proposes a blockchain-based carbon emission rights verification system to further learn proven data using the governance system analysis and blockchain (dApp) mainnet engine to solve these problems.

## 3. UN SDGs' Performance and Blockchain Algorithms for Design and Implementation

### 3.1. Issue Raising

Blockchain is also called a public exchange book, and it is not a central manager that manages the transaction book, but is managed instead by all traders together without a central manager. According to the current banking system, all trading books are owned by banks and can be traded only through banks, which can cause major problems in the event of hacking. In contrast, blockchain does not need a central management system, and it is safe from hacking, as all traders have their own transaction books and their books are constantly updated. With the Paris agreement taking effect, carbon emission rights are one of the hottest issues. China recently announced the implementation of the emission

trading system in the power generation sector, and Korea is considering linking the emission market with China and Japan [47,48].

In addition, California and Canada's Quebec have linked emissions trading systems. The European Union has a common emissions trading system. Japan operates a common offset system with 17 developing countries, including Vietnam. In other words, the world is striving to link the carbon markets between countries. This is possible because there is an international agreement called the Paris Agreement, including a common unit of emission calculations called tCO2-eq. In this respect, emissions rights have very similar characteristics to blockchain-based transactions. The maximum permissible amount of greenhouse gas emissions across the globe can be calculated according to the Paris Agreement's goal of curbing global warming (2 °C). In addition, based on each country's greenhouse gas reduction targets, the maximum amount of emissions allowed by the Earth can be defined in the future. The Paris Agreement also applies the same standards to developed and developing countries. In addition, if the maximum greenhouse gas emission capacity of the Earth or the emission rights under the national reduction objective are allocated to the country and businesses, the emission rights can be traded on the same basis with a common worldwide transaction unit: Carbon dioxide equivalent. The Paris Agreement strictly requires prevention of double counting of emissions (reduction) through transparency mechanisms. Blockchain has the advantage of lowering transaction costs and ensuring transparency, as it is managed through common trading accounts around the world. Last year saw the world's first carbon emission trading between a Russian carbon fund and an African carbon emission holding company. In addition, emissions trading using blockchain was a key issue at the International Association of Emissions Trading in Barcelona. The era of the single global emission trading market and emission currency using blockchain could come faster than expected. Blockchain is also a key topic for the fourth revolution in state affairs. Although the UN and others are talking about the possibility of P2P transactions, we are proposing blockchain governance based on blockchain performance and stability through the main algorithms of blockchain.

*3.2. Research Methodology*

Governance is the generic term for the actions of all decision-making processes that create, update, and discard the formal and informal rules of a system. It consists of factors such as rules, smart tradeoffs, laws (punishment for malicious actors), procedures (what will be done when X occurs), or responsibilities (as to who should do what). The types of governance are largely blockchain governance, project governance, and fund governance. Blockchain governance also proceeds in the form of voting. Of course, voting can be used not only in voting, but also in other ways; also in terms of efficiency, voting is commonly used. The most representative blockchain determines the direction of the network through BIP (Bitcoin Improvement Proposals), EIP (Ethereum Improvement Proposal), developer forums, and various communities; in addition, various blockchains manage the network in slightly different ways. The meaning of project governance is that blockchain networks are a world. Based on this world, we can share on-chain governance and off-chain governance. On-chain governance, the former, refers to a form in which governance can be achieved within a chain without any other elements other than blockchain network components, and blockchain networks can be considered complete as an independent world. Conversely, if governance is done outside the network and it has a direct impact on the blockchain network, this form of governance can be said to be off-chain governance. Fund governance is an act to create a fund to raise the resources needed to create a blockchain ecosystem. Governance is important in blockchain because, over time, the network of blockchain cannot survive without the technology of change that can flexibly cope with the new problems. These quick updates make enterprise and mass market end-user use cases possible. If it takes too long to fix the problem, users will not abandon the service or even participate in it from scratch. Moreover, too-frequent changes divide the community and cause uncertainty to prevail. In a company, a representative's word is law. Thus, changes within a company may be very quick and efficient depending on the capabilities of the representative. In decentralized systems such as blockchain, however, each party has different

incentives. Users, miners, and developers all have different goals. For example, developers were satisfied with Monero's ASIC (Application Specific Integrated Circuit) protection updates, but minor players using ASICs were unhappy, as their profits were shrinking. Eventually, a hard fork broke out, and Monero was divided into several versions. The most important thing in understanding governance is that the incentives that each party constituting governance has may not match. It can be good for users but may be bad for miners (operating companies that implement governance), good for developers but bad for miners. Blockchain governance is designed to achieve network mining agreements through special algorithms. Typical algorithms include Proof-of-Work (PoW), Proof-of-Stake (PoS), and a mix of them. Nonetheless, this governance framework is currently developed and studied extensively, and it has many kinds of governance systems. Funding governance regulates how projects manage the collected funds. Project governance is a kind of meta or meta-meta governance that regulates technology, blockchain governance, funding governance, and meta governance (for example, changing governance procedures altogether). Technology topics include blockchain parameters (block size or gas prices), bug fixes, and new features. On-chain governance is where governance takes place within a blockchain. Ongoing governance involves developers/users/miners alike, and it is largely voted on to modify policies/policy. Off-chain governance is an architecture wherein discussions outside the blockchain are reflected to the blockchain. For example, developers run forums in places like Reddit in a blockchain park, measuring amendments and forming public opinion, and then reflecting them within the blockchain. In cross-chain projects, multiple governance can be implemented simultaneously. For example, in a project called Cosmos, malicious changes cannot be avoided, and rollback procedures must be possible. Blockchain's own token is not a useful incentive or a perfect solution to encourage good behavior. In particular, in situations such as trade shoulder, the token itself can be downgraded as an incentive. In cases like liquid democracy, the centralization of voting rights can be eased but still cannot completely block the populist phenomenon. Because governance is inherently the establishment of mechanisms that address changes in a useful way, several matters are important. In off-chain governance, governance is conducted through online forums outside the blockchain. Developers use the forum to check the revision proposals and their opinions and reflect them as policy changes within the blockchain. Bitcoin mainly runs forums through Reddit. In the case of Etherium, the revised proposals are used to collect public opinion by publishing them on YouTube in lecture form. As such, this paper proposes a Hybrid Governance Protocol, which integrates different types of governance systems. We describe the architecture for introducing blockchain for MRV (Measurement, Reporting, and Verification) application in the GHG (Greenhouse Gas) reduction process. It provides a basic foundation for MRV according to the national CDM (Clean Development Mechanism) methodology. In addition, most nations are developing and equipped with a greenhouse gas reduction zone. There are a lot of data every year, including statistics on greenhouse gas emissions. However, this is because it is the basis for each country to manipulate MRVs. Thus, in order to reduce this GHG, the reduction target must be achieved. There is a lot of discussion about this reduction target. So far, however, the state has arbitrarily granted carbon credits. In addition, each country's guidelines exist, but no clear standardization is made. Thus, the Climate Change and Energy Program emerged as a program to manage climate change and energy. However, these programs are also being challenged by manipulation or by malicious hackers.

Therefore, this research was conducted with Blockchain Agreement Algorithm and Artificial Intelligence Deep Learning Engine for research through verification of GHG reduction and MRV.

In addition, the types of governance structures are largely divided into blockchain governance, project governance, and fund governance.

(1)　Blockchain Governance Structure

This refers to an algorithm that verifies accurate data and various data using blockchain. This level of governance is the stage in which all blockchain governance systems are established. It is also in the process of utilizing existing verification algorithms.

(2)　　Structural Structure Project

The project governance structure is used to establish the ICT (Information Communication Technology) and environmental governance systems required to build and implement carbon emission reduction projects. The project for carbon reduction is a very important step and is related to blockchain agreement algorithms.

(3)　　Fund Governance Structure

Fund governance refers to the blockchain fund governance structure that is needed when the financial sector funds are made when the carbon emission exchange is formed in future P2P transactions in the financial market of carbon emission rights.

### 3.3. Blockchain Process for Reducing GHG and MRV

The flexible mechanism for securing liquidity in the greenhouse gas emission trading market allows allocated companies to use emission rights in various and flexible ways in addition to trading emission rights when submitting emission rights in accordance with their target monthly greenhouse gas emission reduction activities. The purpose of flexibility is to secure liquidity in the emission market by diversifying greenhouse gas reduction methods and inducing more greenhouse gas reduction activities by ensuring flexibility in the method of submitting emission rights. Types of flexible mechanisms include banking and borrowing, recognition of early reduction performance, and certification of external reduction projects, such as offset. In addition, to reduce this MRV, a verification platform that is interlinked with these policy segments must be a mix of goods. This is the same for the whole world. This paper used an artificial intelligence blockchain for the verification platform in line with the process of GHG reduction and MRV reduction, and is expected to show a lot of trading volume if the carbon emission trading system comes out as an individual trading system in the future. Therefore, this paper is equipped with a process to perform these verification procedures. The process required for it is shown in Figure 4.

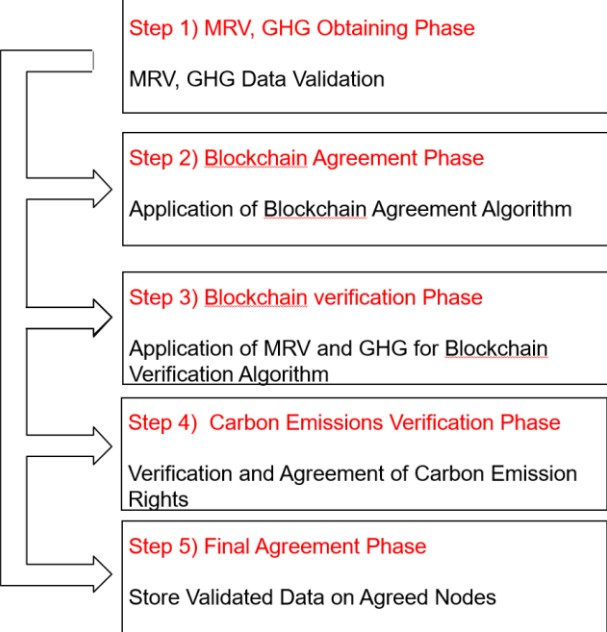

**Figure 4.** Verification blockchain for measurement, reporting, and verification (MRV) and greenhouse gas (GHG) data.

(1)    Step 1

Large amounts of carbon credits will be created, and individuals will be able to purchase them in the future. Thus, the verification of the data is required to reduce the MRV and GHG required.

(2)    Step 2

Blockchain is a step to propose a blockchain architecture that can achieve a very fast agreement according to Hyper PoR (Proof of Random) consensus algorithm and verify it by these consensus algorithms.

(3)    Step 3

The verification algorithm of the blockchain is used to verify each node and to verify it through work in the mesh network.

(4)    Step 4

Algorithms to agree and verify carbon emission rights are obtained. Carbon emission rights are the steps to verifying carbon credits mined by individuals and exchanges necessary for this reduction.

(5)    Step 5

The final validated data divided into agreed nodes after the data consistency verification process are stored. These storage steps are stored thoroughly prepared for hacking.

*3.4. Protocol Design for Hybrid Governance*

The Hybrid Governance Protocol is a transaction that occurs on a blockchain network. Transactions that are recorded on the chain are permanent; thus, once they are written in the network, they can never be erased. Still, many problems can arise from relying solely on the chain. The first is slow speed. After a transaction occurs, it takes quite a long time to be confirmed, so it is difficult to process quickly. The second is privacy. In the case of on-chain, there is a problem wherein the transaction history occurring in the blockchain network is disclosed to everyone. The third is cost and scalability. The cost of all the activities occurring in the blockchain network must be paid for by each person, and the slow speed of transactions is a disadvantage.

Blockchain governance protocols are also a concept created to solve the problems in these on- and off-chains. These protocols can be said to be divided into on- and off-chain protocols. On-chain protocols are those other than blockchain network components, a form wherein governance can be achieved in internal strife without a dispute. Under these special circumstances, blockchain is complete in one independent world. In addition, if this is done outside the governance network, and it directly affects the interior of the blockchain network, then this type of protocol should be designed. In other words, it can be defined as the ongoing governance protocol. Still, this governance is very important because the governance protocol of blockchain plays a very important role in designing consensus algorithms. These protocols have a very important role to play. The SDGs announced by the United Nations have 17 categories. This protocol is also characterized by being optimized for carbon emission rights verification.

These protocols can be developed in the same way as governance systems. This paper compares the Internet environment and the distribution of the value of blockchain by designing these governance protocols and dividing them into application layers and protocol layers, using the text to represent the layer of the network protocol of items in the UN SDGs. The existing application layers, physical layers, and protocol layers are different. This is because the application layer has an accurate stack structure to achieve blockchain decentralization. Moreover, due to the characteristics of blockchain, which is always verified with shared data, the change in the value of blockchain is attributed to

its critical role in the sharing of data. Prior to that, the definition of data is critical. First, existing network protocols define the headers of packets for the data transmission of applications. Transmitters are encapsulated, receivers are encapsulated to create headers for data transmission in protocols, and receivers are referred to as reverse capsules for interpreting and processing headers. Subsequently, hybrid governance can refer to a private–public-based approach to protocol. This meaning will be a breakthrough margin for one-sided protocols of existing single-oriented ones. Submitting blockchain data to a network of applications does not save it all. The protocol makes the agreement, validates the data through the consensus process, and requires approval before the data can be stored. It also tends to be the opposite of the level of applications at which sensitive data are not protected by privacy (see Figure 5).

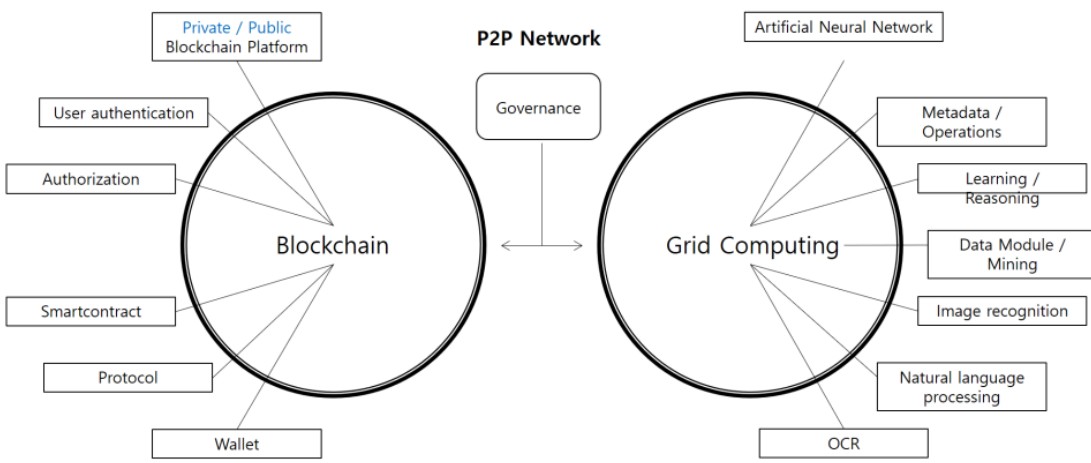

**Figure 5.** Protocol design for hybrid governance.

This meaning is designed to store the desired data whenever a particular event occurs. Modulation or artificial intelligences have a very difficult structure for modulation because of the control of protocols of the data. The hybrid blockchain protocol is designed to be used to verify carbon credits with hybrid structures that go beyond traditional private and public limits.

The task of developing and improving the most important carbon emission blockchain (MRV) must be the central task. Although the government submitted the information specification for the inference of greenhouse gas emissions after receiving third-party verification from the company concerned, the emission trading system will be submitted to the environment minister who is in charge of the emission trading system, and the emission certification committee will review the government certification status following third-party verification. The emission rights can be allocated free of charge, and the proportion of the emission rights allocated free of charge has increased the elasticity of operations by making the global carbon emission rights determined by the Presidential Decree considering the industry's international competitiveness, international trends, and impact on the national economy, such as prices. In other words, emission rights are allocated free of charge by the percentage specified by the allocation plan: 100% during the first planning period, 97% during the second period, and 90% after the third period. It should be noted that Korea's free quota ratio is slightly different from the EU-ETS (EU Emissions Trading System) calculation method. For the EU-ETS, excluding the proportion of emissions allocated on a paid basis is a free allocation ratio; for countries, the percentage of "standard greenhouse gas emissions" of each allocated company is "multiplied by the mandatory reduction rate specified by the company's past average greenhouse gas emissions" to get the free allocation ratio. Therefore, considering the emission rights reserved by the EU-ETS, even though the EU-ETS has a free allocation ratio of 90%, the companies subject to each quota may be allocated all of the standard greenhouse gas emissions free of charge; in Korea's emission trading system, however, only 90% of each allocated company's standard greenhouse gas emissions

are allocated free of charge. Thus, if the allocated company had a similar emission rate in the past, they should purchase 10%.

Ultimately, even if Korea's emission trading system has a free quota ratio of 100%, if 20% of the total emission rights were to be reserved, the EU-ETS ratio would have the same effect as 80% based on the standard. An important factor here is the monitoring service for MRVs using artificial intelligence blockchains. Monitoring means the collection, storage, and management of all data necessary to determine the baseline and to measure artificial emissions and leaks within the business boundaries. Monitoring plans should be established for existing CDM (Clean Development Mechanism) projects and reflected in the project plans. The key elements of the monitoring plan are the data and variables to be monitored, and other data retention deadlines, responsibilities, and authority for data collection and management, quality assurance and management procedures, uncertainty levels, measurement methods, and calibration cycles for measuring equipment should be included; these should include artificial intelligence and blockchain mechanisms. For the data and variables to be monitored, the business plan should include data (variables) names, units of measurement, descriptions of data, data sources, applicable numbers, methods, and procedures, monitoring cycles, quality assurance and management procedures, and the purpose of data use. If sampling is included in the monitoring method, the corresponding sampling plan listed in the project plan is prepared in accordance with the relevant guidance (UNFCCC 2015c), including the sampling design, data to be collected, and implementation plan. The organization of monitoring plans for all CDM projects shall comply with the CDM Business Standards (UNFCCC 2015d). In addition, each approved methodology or new baseline and monitoring methodology will differ in all other details, such as monitoring target, frequency, quality assurance and management procedures, and their levels. Therefore, specific criteria should be established for each item of the monitoring plan in the project plan based on an already approved methodology or a new methodology applied to the CDM project.

Finally, verification of whether such carbon emissions rights are being duly traded is performed based on the CDM verification and certification standards as the procedure for CDM operating bodies to assess the reduction of greenhouse gas emissions by business participants over a specified period of time. The CDM operating body verifies that the project activities comply with the CDM's detailed principles and requirements of the procedure when carrying out the verification. The CDM-certified bodies that perform verification and certification utilize the engine data of the blockchain and present a platform for it considering the grace period and revision of the form.

The above (see Figure 6) stores MRV data, GHG data, and unstructured/structured data in the Web Application Server (WAS). The natural network is mapped to reflect this. These analyzed engines are stored in the Deep Learning repertoire. These stored databases are stored in the real-time enterprise (RTE) server, and the data analyze and deliver databases that can analyze real-time data.

① Transfer MRV data to Web Application Server (WAS).
② GHG data to WAS.
③ Send structured and unstructured data to WAS.
④ Transfer Neural Network data.
⑤ Application of Blockchain Agreement Algorithm.
⑥ Save to Deep Learning data repository.
⑦ Exception Saving on a Processable Server.
⑧ Real-time Enterprise (RTE) is saved to server.
⑨ Real-time DB (DataBase), divided into storage DBs.
⑩ Store on final set database server.
⑪ Display on dashboard.

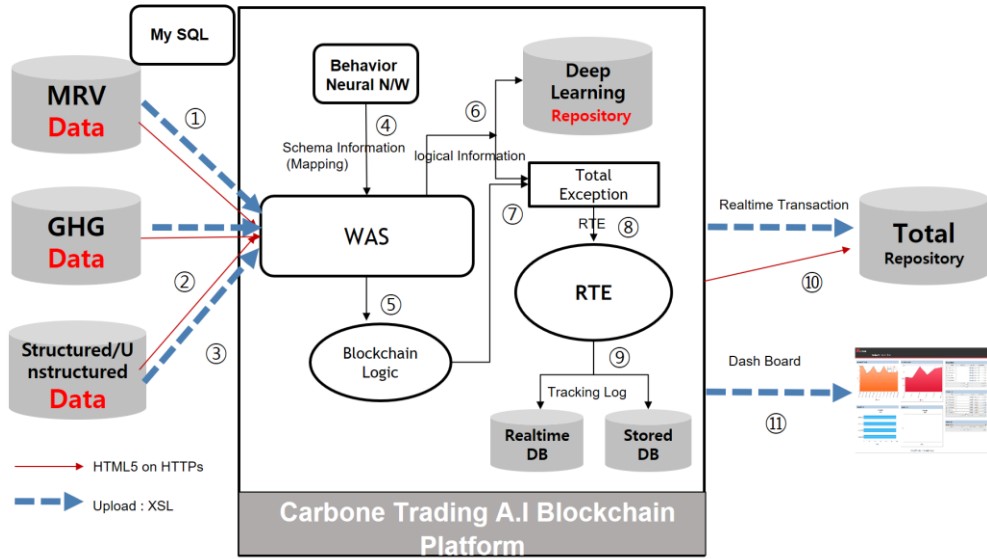

**Figure 6.** Carbon trading AI (Artificial Intelligence) blockchain platform.

## 3.5. Verification Method using Hybrid Governance Protocol

Looking at blockchain-based technology, there are protocols that use the blockchain concept, and the structure is based on protocol-based tokens. Blockchain itself is a distributed database or a data structure. Each node distributes and stores data using encryption technology. The protocol is one that can communicate between blockchain nodes (see Figure 7). Nodes using the same protocol can communicate with each other. An environment where various kinds of software can operate based on the protocol is produced. There are various tokens that operate in such protocol-based environments. Existing blockchains do not provide development environments, such as development environments or virtual machines, so they do not have tokens, but must be accessed in terms of initial value storage. Various kinds of tokens used on the basis of smart contracts can be seen as a program that performs each unique type of contract. Compared to communication, there are protocols for exchanging over the Internet and file/email with protocols such as HTTP, FTP, and SMTP based on TCP/IP technology, and various programs are produced using such protocols. Various programs such as websites, email, and file sharing are kinds of tokens, and there are attempts to create a decentralized world free of intermediaries through blockchain technology.

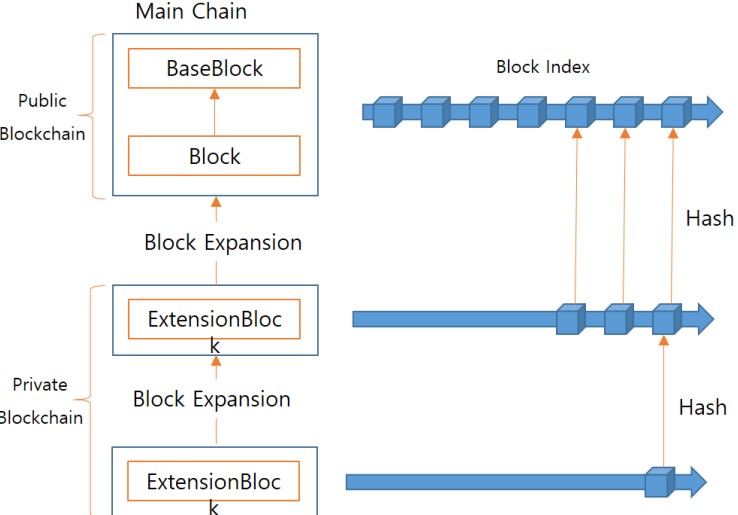

**Figure 7.** Interchain blockchain diagram.

### 3.5.1. Hybrid Governance Protocol Applicable to UN SDGs

A protocol in communication is one wherein a computer or a system communicates with another. Pre-promised operational regulations are required to perform data transfers quickly and accurately and to provide error detection and recovery functions. The Internet has communication over the network based on protocols such as TCP/IP, SMTP, HTTP, and HTTPS. A standard is always necessary, and the network always requires interoperability.

(1) TCP/IP (Transmission Control Protocol/Internet Protocol): A collection of communication protocols used by computers to access the Internet.
(2) HTTP (Hypertext Transfer Protocol): Communication protocol used to exchange documents.
(3) SMTP (Simple Mail Transfer Protocol): Simple e-mail communication protocol used to send and exchange e-mail between computers.

Next, we went through Web 1.0 and 2.0, and on top of these protocols were applications like Google and Facebook. Their business model was to build their own data and, based on that (centralized), demand/charge services directly from customers or collect advertising fees. At that time, the DB was closed, owned. Being able to obtain data and information only by using that service limited the possibility of creating new value from an open source from a different angle. Blockchain leads to a "change in protocol level" that has enabled these existing Web services. With an open-source, distributed DB, anyone can access data and share information freely. It is all about hardware, software, networks, and data. Blockchain protocols allow everyone to download software, become nodes, and replicate and use data. Joel Mongro [49] of Placeholder VC/Union Square Ventures describes blockchain as the embodiment of the Fat Protocol, unlike the traditional web-based Thin Protocol. The value created through the development of protocols and the return on the value created were low because information was locked in the application. Over the next few years, however, blockchain will rebuild protocols in decentralized form, which will revolve around the development and value creation of technologies.

(1) If the data existed in the application layer in the existing web base, blockchain shares the data with anyone on an open source basis.
(2) The underlying protocol project itself can be easily financed by developers through tokenization, and those who develop the dApp on top of that protocol also benefit as the protocol ecosystem grows.

While monetization over protocol was to create and sell S/W (Software) that utilizes the protocol, blockchain allows people who agree to the protocol to purchase each token and create better value for that protocol, resulting in an increase in value and price so that people can immediately obtain economic rewards. Thus, creating a new protocol and becoming dominant create the value of the token, giving rise to the underlying protocol project itself. They share computing power (Data Base), but they can decentralize and manage their resources. Now that the security tokenization is in earnest, we think it will be possible to develop more active protocol levels. In the early days, when one thinks of Netscape and Google, the Web was more like a thin protocol. In fact, data showed that most investments were made in infrastructure corresponding to protocols/core throughout 2017. Project financing in infrastructure amounts to 10 billion USD, with the sum of infra, payment, finance, exchange, and computing, which correspond to protocols and major vertical apps, practically making up the majority. Actually, the return is also high here. In a token economy, all assets are tokenized, and people take advantage of the goods/services written on top of the protocols they support, purchase token/coins in the process, and, in return, get rewarded as much as all participants contribute. However, we have a long way to go, and we do not think that human-developed technology itself will solve the social problems that we care about. On the other hand, if we remain still for many reasons, the world of blockchain and decentralization will not be as beautiful as expected [50].

Certificate-Based Hybrid Blockchain

The certificate-based hybrid blockchain's public key password is used to authenticate the certificate based on the certificate containing the information of the certificate. In Korea, however, the Korean government has prepared a system for issuing public certificates through the 1999 Autobiography Act to certify the certificates through five public certificates under the best certificate authority, Root CA. For foreign countries, personal in-device, Eve modem device authentication, and WiMAX industry certificates are provided through Verisign's device certification service. In addition, the certificate-based authentication technology is being used in VoIP (Voice over Internet Protocol) and network surveillance cameras, and the station is expanding. Certificate-based authentication technology provides high stability through strong authentication functions and supports anti-denial functions. Nonetheless, device certification processing software algorithm requires high computational throughput. Therefore, it is not suitable for use in IoT (Internet of Things) devices that have power and performance. As such, blockchain is used instead of certificates for authentication to present a functional and usable enough blockchain.

Authentication of Hybrid Blockchain based on Cryptographic Protocol

A protocol that authenticates objects based on the private and public blockchains' public key passwords and symmetric key passwords is used primarily in wireless Internet security protocols. It supports various tables, such as 802.1x/802.11i and WPA. The cryptographic protocol-based authentication method can include technologies such as ID/password-based authentication, MAC-address-based authentication, and certificate-based device certificate authentication depending on the cryptographic protocol used. A variety of authentication methods can be provided to select the combined authentication method according to the use environment, and the anti-denial function can be supported according to the code protocol adopted. Depending on the security-based encryption technology, however, if any vulnerability in encryption technology is found, it can lead to vulnerability in authentication technology. These blockchain technologies offer many blockchain architectures based on various encryption technologies.

Certification of Hybrid Blockchain based on MAC Address

MAC address is an authentication method using a unique identification address (MAC) address assigned to the underlying hybrid network interface, which is used primarily for network rooting in intranet environments. It has a difference that compares the MAC address registered in the server with the message sent from the device when the device requests one in the network, and it is easier and faster than the ID/password-based authentication method. With the advent of IoT, however, there is a need for new MAC address forms to be defined, and new tables are being defined. MAC addresses can be tightened, so without extra security equipment, they are vulnerable to attacks by spoofing and others. These hybrid-based blockchain types can be applied to a variety of industry groups for hybrid-based blockchain authentication based on the MAC address of a computer system. In addition, blockchain is proposed for authentication work for authentication and protocol.

Authentication of Hybrid Blockchain based on ID/Password

This is a technology used mainly in server/client authentication environments by storing each user ID and password of blockchain users in the server's DB and authenticating them based on stored knowledge. To prevent a list of passwords stored on a server from being exposed and consequently disabling authentication, a method of storing values through a hash function is often adopted. To ensure greater safety, hide SSIDs (Service Set IDentifier), use the WEP key between AP and device, adopt the PAP authentication method, and use the RFID method. The ID/password method in the IoT environment has problems, such as the server's load due to the characteristics of the IoT environment,

where many devices are used without human intervention, and it is important to exclude human intervention in the additional process of device modification.

The problem is that it is not suitable for authentication technology in an IoT environment because it does not provide an anti-witching function. Based on blockchain technology for these IDs/passwords, it provides the necessary skills for the most basic authentication.

### 3.5.2. Hybrid Governance Protocol for Design Pattern

Visitor patterns are often used with composition patterns. In the picture above, when dealing with the class that implements an element on the left, the composition pattern is used to implement the visitor pattern by implementing the individual objects and their parent objects in the form of a list. In general, NodeBlock allows a member variable and a method to be written together within an object; in a visitor pattern, however, the variable and method are created separately. For a similar pattern, Java does not support multiple inheritances from multiple different objects simultaneously. The term multiple inheritance here means that language internally supports double dispatch, and that multiple inheritance is not supported. The non-identifier pattern eventually results in the object's inheritance to the element and the non-identifier's inheritance, which is not supported by Java, resulting in a somewhat complex form (two interfaces and two implementations). The block is also very important because of the pattern design, which is very important here. AssignmentBlock and VariableRefBlock have many meanings with regard to the status of blockchain.

### Hybrid Governance Protocol for Visitor Pattern

This design is a common way of implementing processing-related logic in each node class when processing data are formed from syntax trees, file systems, or hierarchical structures that are often encountered among design patterns. The design patterns nodeblock and typecheckingblock can be found in a diagram of the file system in the class and coxegeneratingblock. Recursively, the methods have a logic output, i.e., a list of files. This is called, like Linux, ls-R, which is lower in the output to the list of files. An abstract method named to the abstract class from a higher node, which probably looks solved, is added by the respective method in classes and implementation. Since these requirements increase the number of nodes or the number of logic units handling each node, the analysis should be done by looking at the logic that is sweeping around the code. Moreover, the difficulty of debugging or analyzing related logic increases because of the lack of coordination of related logic; if there is a common task in some processing, the common tasks are spread across all nodes, and they can lead to shotgun surgery if changed. To process the cohesive logic and common logic as much possible, assignmentblock is extracted and used separately. When files are moved or copied to a specific location while traveling around a file tree, the code may have asynchronous methods that are passed to a separate system other than variableRefBlock, or the code is highly cohesive, which increases reuse and allows extended processing of node classes without modification, thus complying with the blockchain principle. On each node directory path in particular, the actual data are calculated to send a webrequest (see Figure 8).

### Hybrid Governance Protocol for Observer Pattern

Using the interface, the Observer pattern of the Hybrid Governance Protocol recommended loose coupling between objects. In other words, it uses composition, not implementation through inheritance. Configuration means that A has B, which is often used to include interfaces rather than objects in an object. The Observer pattern that we are trying to use also uses interfaces like this. There are two main roles to play here. One is the role of Publisher, and the other is the role of Observer. Once they have defined their interfaces, they use the initial classes. The interface called Publisher will have a method of managing Observers. Three methods can be defined: Receiving and registering (add), excluding (delete), and notifying registered Observers (notifyObserver). The Observer interface will have an update method that updates the information. If you draw a class diagram based on this,

it will be shown below. The CodeGenerationVisitor class, which implements the Publisher, becomes the informative Publisher and has Observer objects. Likewise, the Node and NodeBlock classes implementing the Observer interface update are called each time TypeCheckingVisitor notifies you when calling a new Node. Based on this, let us code it as JAVA. The way Publisher sends information or status changes to the Observer is called push, as we commonly know it. On the other hand, the way the Observer asks the Publisher for information whenever it is needed is called pull. So far, it has been a push-button approach. Of course, a pooling method can be implemented with an observer pattern. In JAVA, we basically provide the applications of this Observer pattern. This is the Observable class and the Observer interface. This makes it easier to apply Observer patterns without having to implement a direct interface. Nonetheless, there is one problem here, as there are pros and cons. Specifically, Observable is not an interface, but a class. In the end, if you are forced to inherit, but the inheritance must be from another class, you cannot use it. Because JAVA does not support multiple over-the-counter inheritances, it will need to be used appropriately in view of the circumstances (see Figure 9).

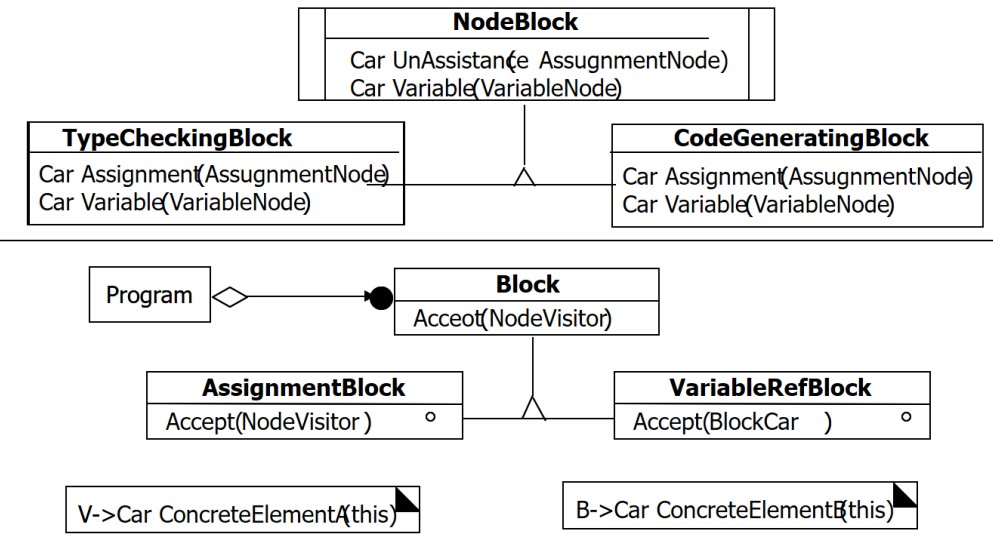

**Figure 8.** Hybrid Governance Protocol for the design pattern.

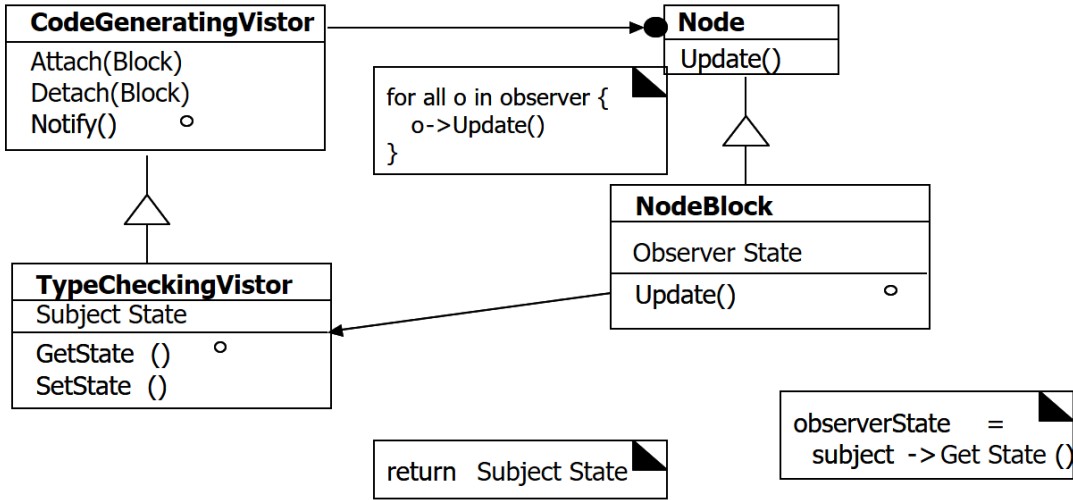

**Figure 9.** Hybrid Governance Protocol for Observer pattern.

### 3.5.3. Implementation of the Hybrid Governance Protocol

This is the first time to download and run for Hybrid Governance. It must be connected to some nodes to download the latest blockchain. This refers to the relationship between these nodes, considering the fact that the computer does not recognize all or some nodes in Bitcoin. Hardcoding of node addresses in Hybrid Governance can be incorrect. The node may be attacked or shut down, preventing the new node from participating in the network. In Hybrid Governance, the DNS (Domain Name System) seeds are hard-coded instead. This is not a node, but a DNS server that knows the addresses of some nodes. The first time you run Hybrid Governance, the core will be connected to one of the seeds, and you will get a list of addresses of the pool nodes to download the blockchain.

In our implementation, however, centralization will be achieved. We have three kinds of nodes.

(1) Central node: A node connected to all nodes and a node that transmits data between different nodes.
(2) Collector node: This node stores new transactions in mempool and mines new blocks when enough transactions are stacked.
(3) Mobile node: This node is used to transfer coin between wallets. Unlike the SPV (Simple Payment Verification) node, it stores a full copy of the blockchain.

Below is the target scenario.

(1) Central nodes create a blockchain.
(2) The other wallet nodes are connected to the central node, and the blockchain is downloaded.
(3) Collector nodes are also connected to the central node to download the blockchain.
(4) The wallet nodes generate transactions.
(5) The collator node receives the transaction and stores it in the memory pool.
(6) When enough transactions are accumulated in the memory pool, the miner node begins mining the new blocks.
(7) When a new block is mined, it is sent to the central node.
(8) The wallet node is synchronized with the central node.
(9) The user of the wallet node checks if payment is made normally.

The node communicates through the message. When a new node runs, it imports multiple nodes from the DNS seed and sends a version message (see Figure 10). The message can be implemented as follows. The Version field is not important because our blockchain has only one version. BestHeight stores the length of blockchain that a node has. AddrFrom stores the sender's address. The node that receives the version message responds with its version message. This is a kind of handshake, and we cannot interact until we send and receive greetings from each other. This is not simply an expression of politeness. Version is used to find a longer blockchain. Upon receiving a version message, a node checks if its blockchain is longer than BestHeight. In shorter cases, the node requests and downloads the missing blocks. You need a server to receive a message (see Figure 11).

```
type version struct {
        Version     int
        BestHeight  int
        AddrFrom    string
}
```

**Figure 10.** Importing multiple nodes.

```
var nodeAddress string
var knownNodes = []string{"localhost:3000"}

func StartServer(nodeID, minerAddress string) {
        nodeAddress = fmt.Sprintf("localhost:%s", nodeID)
        miningAddress = minerAddress
        ln, err := net.Listen(protocol, nodeAddress)
        defer ln.Close()

        bc := NewBlockchain(nodeID)

        if nodeAddress != knownNodes[0] {
                sendVersion(knownNodes[0], bc)
        }

        for {
                conn, err := ln.Accept()
                go handleConnection(conn, bc)
        }
}
```

**Figure 11.** AddrFrom stores the sender's address.

First is the hard-coding of the address of the central node. All nodes need to know which node to connect to at first (see Figure 12). The minerAddress factor specifies the address where mining compensation will be received. If the current node is not a central node, then a version message must be sent to the central node to ensure that its blockchain is up to date. The message is a sequence of bytes viewed at the low level. The first 12 bytes represent the command name ("version") followed by the gob-encoded message structure. CommandToBytes is implemented as follows:

```
func sendVersion(addr string, bc *Blockchain) {
        bestHeight := bc.GetBestHeight()
        payload := gobEncode(version{nodeVersion, bestHeight, nodeAddress})

        request := append(commandToBytes("version"), payload...)
        sendData(addr, request)
}
```

**Figure 12.** Carbon node to connect.

This function creates a buffer of 12 bytes, populates the command name, and leaves the remaining bytes blank. Conversely, there is also a function of converting a byte sequence into a command. Upon receiving a command, a node takes the command name through bytesToCommand and processes the command content with the appropriate handler. First, decode the request and import the payload. Since this is common to all handlers, this code will be omitted from the code snippet to be written in the future. The node then compares its bestHeight with the value received in the message. If the blockchain of a node is longer, reply with a version message; if it is shorter, send the getblocks message (see Figure 13).

When the block hash values are received, they are stored in the blocksInTransit variable to track the downloaded blocks. This lets you download blocks from other nodes. Immediately after switching the block to the transport state, send the getdata command to the node that sent the inv message and update blockInTransit. In the actual P2P network, you try to transfer blocks from different nodes, not just from the node that sent the message. Our implementation does not transmit multiple hashes

to Inv. This is why the first hash is only taken in payload. Type = "tx". Check if there is a hash in the mempool; if there is no hash, send getdata.

```go
func handleVersion(request []byte, bc *Blockchain) {
        var buff bytes.Buffer
        var payload version

        buff.Write(request[commandLength:])
        dec := gob.NewDecoder(&buff)
        err := dec.Decode(&payload)

        myBestHeight := bc.GetBestHeight()
        foreignerBestHeight := payload.BestHeight

        if myBestHeight < foreignerBestHeight {
                sendGetBlocks(payload.AddrFrom)
        } else if myBestHeight > foreignerBestHeight {
                sendVersion(payload.AddrFrom, bc)
        }

        if !nodeIsKnown(payload.AddrFrom) {
                knownNodes = append(knownNodes, payload.AddrFrom)
        }
}
```

**Figure 13.** Getblocks message code.

## 4. Experimental Results

### 4.1. Experimental Environment

It is necessary to verify through a test net to develop dynamic (within one second of overhead) multi-channel blockchain technology that supports real-time transaction parallel processing through graph profiling of carbon emission blockchain event-based health management.

Establish an environment to obtain blockchain performance data.

System environment

- Server: HP ML-500-6
- Network: 5G Bps
- Environment: AWS Clode
- Node: 100
- Database: IPFS

### 4.2. Experimental Conditions

(1) Real-time multi-channel communications protocol.
(2) Improve transaction parallelism using profile information and speculative execution.
(3) Event-based multi-chain health management system development.
(4) Hierarchical block chain creation and maintenance system.
(5) Parallel processing and optimization of transactions using dependent graphs.
(6) Develop dynamic channel allocation and merge technologies.

Finally, carbon verification is required among carbon emission blockchain verification systems based on the end point.

(1)    Major Functions

-       Multi-chain health management system.
-       Multi-chain block creation and maintenance system.
-       Multi-channel communication protocol module.
-       Virtual machines that process transactions in parallel through profiling and speculative execution.
-       Dynamic channel allocation and merging algorithm.
-       Blockchain test net with scalability solution for this task.
-       Mainnet (SW).

(2)    Major skills

-       Blockchain-based NoSQL real-time database and health path event function structure.
-       Transaction separation processing system based on channel-to-channel dependencies.
-       Blockchain virtual machines that can execute speculation through parallel processing and profiling of transactions in a channel.
-       A system that dynamically manages channels using profile information to ensure uniform transaction throughput for each channel.

(3)    Comparison of key performance data

-       Transaction processing speed 15,000 TPS.
-       One hundred concurrent transaction types.
-       Thirty seconds of throughput for transactions dependent on more than one channel.
-       Ten times the in-channel transaction parallel throughput.
-       Overhead of two seconds for dynamic channel management.

*4.3. Blockchain Validation Comparison*

This measures the time that it takes for transactions to process the change that the state managed by more than one channel for verification of carbon blockchain. It also compares the transaction throughput (TPS) measured in parallel with the transaction throughput (TPS) in the control group, which continuously processes the transactions within. Figure 14 shows the average value of 12,000 averages when these performance data are rotated five times.

In addition, the testbed network environment consisting of 10 nodes is established, and the performance data are measured with approximately 15,000 TPS performance data values when measured, as shown in Figure 15, and about 100 times when the test bed network environment is constructed with the same 10 nodes as the client sending the transaction request to each channel through a non-numeric method. Thus, the results of measuring the number of channels that can be generated while maintaining the appropriate number of transactions in a testbed network environment consisting of these 50 identical nodes show that the performance information is stable as the average value is paid 100 times.

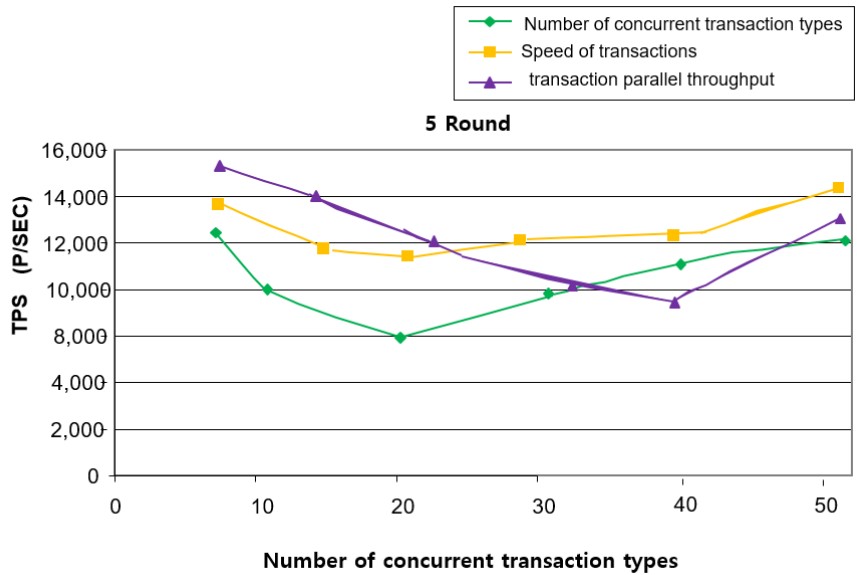

**Figure 14.** Number of current transaction types of performance data (average number 5).

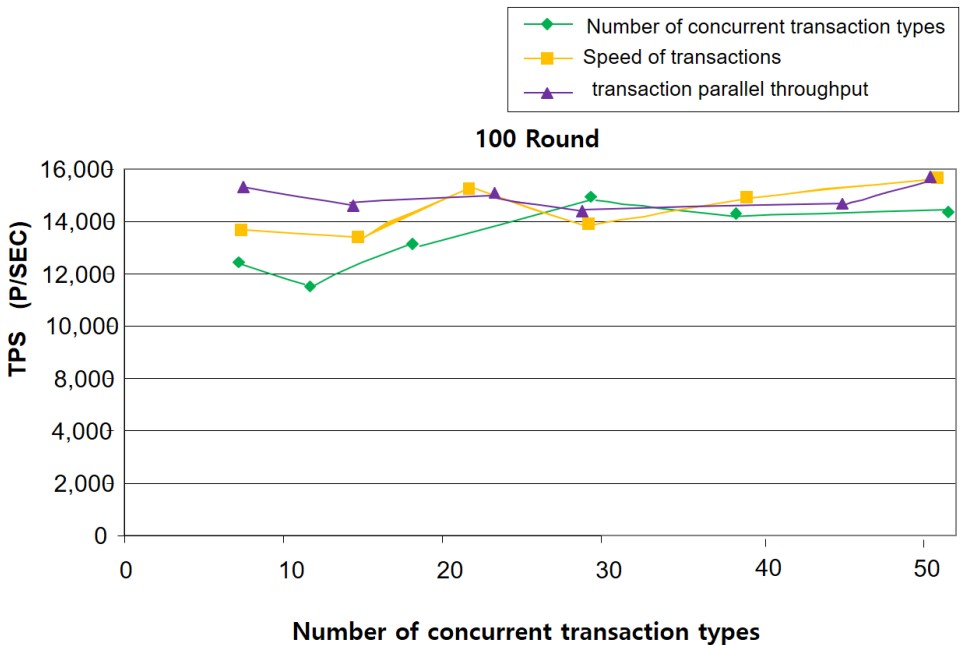

**Figure 15.** Number of current transaction types of performance data (average number 100).

## 5. Discussion

Discussions have been held since 2017, and the consensus is that it is not too early to modify the system, but it was concluded in August that Bitcoin cache is hard to fork out to networks. Likewise, a problem exists in the trade of carbon credits according to UN-SDGs between individuals. Because the governance system became very important in validating carbon credits with peer-to-peer (P2P), the governance system is applied to carbon emission rights based on the stability, decentralization, and reliability of the blockchain. This created the dApp, which developed the carbon emission blockchain. The ID for certification of these carbon credits, PWD (Password), will be used to certify these carbon credits. Still, there is one problem that needs to be solved in a decentralized network. While start-ups and other general companies can quickly modify and put products on the market (especially if the product is software), blockchain networks are decentralized, making it very difficult for anyone to

respond quickly to what the market wants. That is why the way of making decisions in such a decentralized network becomes very important. In addition, once the network starts, there will be a number of stakeholders, including miners (or witnesses and validators), developers, coin holders, and coin investors, all of whom have different interests, thereby making it quite difficult to coordinate and reach an agreement. In addition, in some cases, the security and stability of the network are more important than quick response to the market, such as with Bitcoin, so it is important to think about how to make decisions when designing a network. Therefore, governance becomes very important because decision-making is not effective in a decentralized network, or it can be a major impediment to the growth of the network if it flows in the wrong direction.

In addition, the required dApp is selected by choosing either carbon credits or energy blockchains. These decisions are made, and the energy agreement algorithm-applied data are selected. The members can calculate the usage based on the use of carbon emission using various points, and so on (see Figure 16). Then, the required dApp is selected to run the energy blockchain.

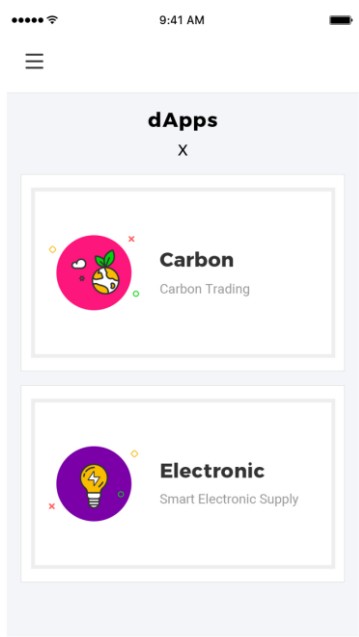

**Figure 16.** Carbon emission blockchain login screen.

One must decide between the actual seller and buyer and click the add button to build and operate an applicable dApp for the carbon emission blockchain (see Figure 17).

One can also choose the type of carbon emission that one needs. For example, one can choose from solar energy, wind energy, or geothermal energy and actually buy the power one needs (see Figure 18).

This paper followed the recommendations of the UN-SDGs, and there were many problems in dealing with individual carbon credits in the existing carbon emission system. The first reason is that the country is now centralized in carbon trading. Second, in order to conduct turn-on transactions among individuals, authorized agencies must measure and verify carbon transactions. Nonetheless, it is difficult to verify all of them centrally in an individual transaction. Therefore, the blockchain's artificial intelligence decentralization service has been designed.

In addition, the necessary dApp was introduced. In fact, we can protect against carbon emissions anomalies by using big data and artificial intelligence in mobile cloud environments. In addition, there is no way of restoring the private key used in electronic signatures when using blockchain in carbon emission monitoring/trading using smart collect, or of protecting the private key from being hacked (see Figure 19). As such, it is necessary to establish criteria for verifying the normal operation of smart contracts because programs recorded in programs can operate abnormally, causing problems

such as economic damage or personal information leaks. One will have to search in advance. To solve these problems, this study proposed a blockchain-based carbon emission rights verification system to learn proven data further using governance system analysis and a blockchain mainnet engine.

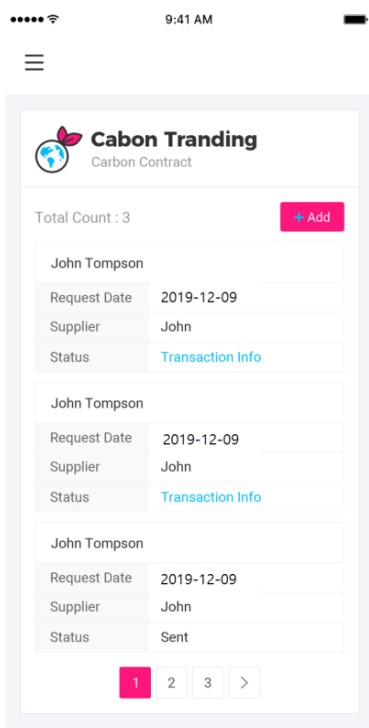

**Figure 17.** Carbon emission blockchain trading screen.

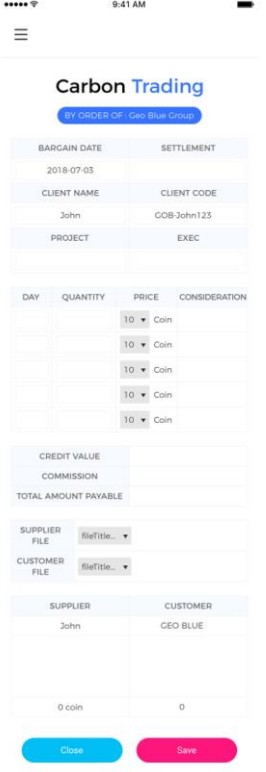

**Figure 18.** Details of a carbon blockchain transaction.

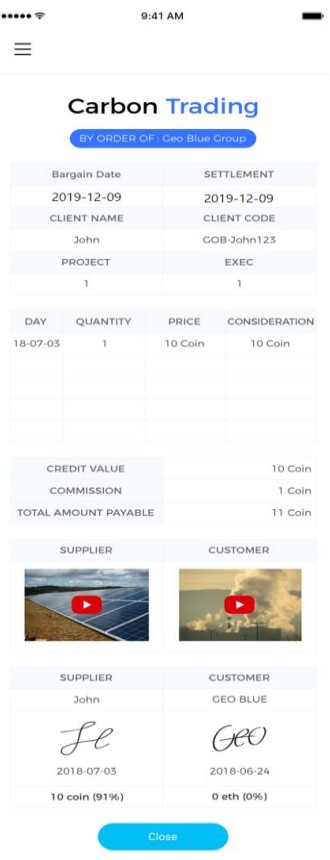

**Figure 19.** Carbon blockchain monitoring/trading system.

## 6. Conclusions

While many topics, such as consensus algorithm (PoW, PoS, etc.), coin economy (how to distribute coin/talk to whom, etc.), and scalability, are discussed in the blockchain-based decentralized network, governance related to how to make decisions is not likely to be discussed well. Thus, in this paper, the centralized network talks about why the UN-SDGs' governance is important. In general, planning a product is a process of looking at what the user's potential demand in the market is, and then designing a solution that can meet this demand (i.e., addressing the user's problems). Even if one identifies the hidden needs of the market, it is very difficult to create a product that can accurately address those needs, i.e., "shooting at what the market wants." In most cases, no matter how well prepared and released the product is, it does not capture the demand of the market fully, but meets it only partially. Because it is so difficult to predict demand in the market, instead of planning elaborately at first with high cost and effort, the company makes a minimal functional product (MVP) that can meet the expected market demand even if quality drops a little, and then quickly modifies the product according to the market response to find a product–market combination at the beginning of the business. This need to make products that satisfy what the market wants is the same for decentralized projects based on blockchain. It is equally difficult to satisfy the market, so no matter how sophisticated the blockchain characteristics and token explosions are before the start, the market rarely flows in the direction predicted before the launch of the network; in most cases, the market will flow in a different direction from the originally predicted one. It would be the same for blockchain-based decentralized projects that have to make products satisfying the needs of the market, but it is not easy to satisfy the market; thus, no matter how sophisticated the blockchain characteristics and token explosions are before the start, the market rarely flows in the direction predicted before the launch of the network; in most cases, the market runs in a different direction from the originally predicted one. Carbon credits are currently a very difficult technology to physically verify with emission rights. Therefore, this paper aims to establish

a governance system for the UN-SDGs to verify these carbon emissions. In order to establish this governance system, technological independence must be achieved first. Technology is something that needs to be reliable. As such, this paper designed the architecture to enhance the performance information of the trust-based blockchain. It is also equipped with an expanded version of the DPoS (Delegation Proof of Stake) method, an agreement algorithm. Second, the verification encryption algorithm needed to build the engine of the blockchain is discussed. The verification algorithm is a performance- and trust-based encryption algorithm. The third point is the convergence of artificial intelligence and blockchain algorithms. It discusses the mechanism of using blockchain and artificial intelligence algorithms to make sure that carbon emission rights are the correct or wrong answer. This paper now says that carbon credits were previously converted into B2B (Business-to-Business) markets from carbon emission exchanges, but are now expanding to P2P markets. Therefore, if this P2P (Peer-to-Peer) market becomes a P2P market, this is a paper for establishing and verifying carbon emission blockchain governance to verify carbon emission rights.

**Author Contributions:** Conceptualization, S.-K.K. and J.-H.H.; Data curation, S.-K.K.; Formal analysis, S.-K.K.; Funding acquisition, J.-H.H.; Investigation, S.-K.K. and J.-H.H.; Methodology, S.-K.K. and J.-H.H.; Project administration, J.-H.H.; Resources, S.-K.K. and J.-H.H.; Software, S.-K.K. and J.-H.H.; Supervision, J.-H.H.; Validation, S.-K.K. and J.-H.H.; Visualization, J.-H.H.; Writing—original draft, S.-K.K. and J.-H.H.; Writing—review and editing, J.-H.H. All authors have read and agreed to the published version of the manuscript.

**Funding:** This work was supported by the National Research Foundation of Korea (NRF) grant funded by the Korean government (MSIT) (No. 2017R1C1B5077157). In addition, this research was supported by the Energy Cloud R&D Program through the National Research Foundation of Korea (NRF) funded by the Ministry of Science, ICT (NRF-2019M3F2A1073385).

**Conflicts of Interest:** The authors declare no conflict of interest.

## Abbreviations

| | |
|---|---|
| SDGs | Sustainable Development Goals |
| ET | Emission Trading |
| JI | Joint Implementation System |
| CDM | Clean Development Mechanism |
| OTC | Over-the-Counter |
| UN | United Nations |
| EU | European Union |
| TPS | Transactions Per Second |
| P2P | Peer-to-Peer |
| IT | Information Technology |
| MDGs | Millennium Development Goals |
| CDM | Clean Development Mechanism |
| KOC | Korean Offset Credit |
| KCU | Korean Credit Unit |
| KAU | Korean Allowance Unit |
| EU-ETS | EU Emissions Trading System |
| MRV | Measurement, Reporting and Verification |
| MAC | Media Access Control Address |
| BIP | Bitcoin Improvement Proposals |
| EIP | Ethereum Improvement Proposal |
| ASIC | Application Specific Integrated Circuit |
| PoW | Proof-of-Work |
| PoS | Proof-of-Stake |
| GHG | Greenhouse Gas |
| CDM | Clean Development Mechanism |
| ICT | Information Communication Technology |
| PoR | Proof of Random |

DB          DataBase
AI          Artificial Intelligence
S/W         Software
VoIP        Voice over Internet Protocol
IoT         Internet of Things
SSIDs       Service Set IDentifier
DNS         Domain Name System
SPV         Simple Payment Verification
MRV         Measurement, Reporting and Verification
WAS         Web Application Server
TCP         Transmission Control Protocol
IP          Internet Protocol
DPoS        Delegation Proof of Stake
B2B         Business-to-Business
PWD         Password
HTTP        Hypertext Transfer Protocol
SMTP        Simple Mail Transfer Protocol

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
