# Peer review of "Blockchain of Carbon Trading for UN Sustainable Development Goals"

_sustainability, doi:10.3390/su12104021_

Round 1

Reviewer 1 Report

The authors propose the use of the blockchain technology for carbon trading. 

I think that some parts of the paper are badly organized and not well written. Some paragraphs are confusing and/or contain unnecessary amount of information. A scientific article should be instead clear and concise.

In the following, I show some aspects, which must be necessarily improved:

1. At the end of Section 1, I would like to see a paragraph, which introduces the rest of the paper: "The remainder of this paper is organized as follows: Section 2 introduces... ; in Section 3 we show..., etc." This could give more order to the manuscript and make the reader more aware of what he/she will read on the following pages.

2. Acronyms must be properly defined before being used. Not every reader is an expert. Some examples are listed below:

  • line 20: "UN-SDG" 
  • line 54: "OTC"
  • line 55: "SDG"
  • line 57: "UN"
  • line 94: "CDM"

Please, observe that there are many other cases in the paper.

3. Figures should be described and explained in detail. The elements of Figure 1 are not explained within the paper. You only write "see figure 1"; for example what Poseidon App is? It is not explained. The acronyms of Figure 2 should be explained (e.g., KAU, KCU, KOC), so that the reader can understand its meaning immediately. The sources of Figures 2-3 need a reference in the bibliography. 

4. The source of Table 1 needs a reference. Moreover, some parts of such table have a different font.

5. At the beginning of Section 2 you show a list of papers published about a certain topic. It is unnecessary, because the reader can retrieve this information from the bibliography. Moreover, you cite 4 papers, but the references reported at the end of the paragraph are only 3 and wrong "[3-5]". Please, check.  

6. The references should be written according to the order of appearance in the paper. Please, see [36-38] (line 140). Please, check the overall paper.

7. The subsections 2.2.1-2.2.6 need references, especially as regards the numerical data that are reported.

8. The sentence "The types of governance are largely blockchain governance, project governance, and fund governance." needs to be better explained. 

9. Figure 5 is well-built and interesting, but it should be better explained in the paper. Which is the flow? The mechanism? How do the various parts interact with each other?

10. In line 524, you talk about Joel Mongro; a reference is needed.

11. The conclusions should only clarify and summarize the main aspects of the article. Therefore, I think that Figures 13-16 should be placed and explained before starting the conclusions. 

12. Your contribution must be clearer. Are there articles in the literature that deal with the use of blockchain technology in the carbon trading? What is introduced differently in your paper? There are some papers in the literature about this topic. A brief comparison should improve the overall quality of your manuscript. Some examples are:

  • Pan et al. (2019), Application of Blockchain in Carbon Trading, Energy Procedia;
  • Khaqqi et al. (2018), Incorporating seller/buyer reputation-based system in blockchain-enabled emission trading application, Applied Energy.

These are just some examples, you don't necessarily have to mention these.

13. Is there experimental evidence about the effectiveness of dAPP? Please, deepen this aspect.

Author Response

Comments and Suggestions for Authors

The authors propose the use of the blockchain technology for carbon trading.

I think that some parts of the paper are badly organized and not well written. Some paragraphs are confusing and/or contain unnecessary amount of information. A scientific article should be instead clear and concise.

In the following, I show some aspects, which must be necessarily improved:

  1. At the end of Section 1, I would like to see a paragraph, which introduces the rest of the paper: "The remainder of this paper is organized as follows: Section 2 introduces... ; in Section 3 we show..., etc." This could give more order to the manuscript and make the reader more aware of what he/she will read on the following pages.

Reply->

Thank you for your valuable comments. I revised the paper as per your recommendation. The revised or added parts are being highlighted in Red/Green for your possible re-review.

Also, we use blockchain to verify carbon emission rights in our efforts to manage energy. These efforts are taking a little further into the verification system for blockchain. But so far, I think the blockchain carbon emission system is in its early stages. Therefore, in order to prevent hacking and strengthen security vulnerabilities, the available blockchain network protocol will be applied and changed to the appropriate blockchain network protocol for the greenhouse gas comprehensive information center.

  1. Acronyms must be properly defined before being used. Not every reader is an expert. Some examples are listed below:

line 20: "UN-SDG"

line 54: "OTC"

line 55: "SDG"

line 57: "UN"

line 94: "CDM"

Please, observe that there are many other cases in the paper.

Reply->

Thank you for your valuable comments. I revised the paper as per your recommendation

Abbreviations

SDG: Sustainable Development Goal

SDGs: Sustainable Development Goals

ET: Emission Trading

JI: Joint Implementation System

CDM: Clean Development Mechanism

OTC: Over-The-Counter

UN: United Nations

EU: European Union

P2P: Peer-to-Peer

MDGs: Millennium Development Goals

CDM: Clean Development Mechanism

KOC: Korean Offset Credit

KAU: Korean Allowance Unit

EU-ETS: EU Emissions Trading System

MRV: Measurement, Reporting and Verification

PoW: Proof-of-Work

PoS: Proof-of-Stake

GHG: Greenhouse Gas

MRV: Measurement, Reporting and Verification

WAS: Web Application Server

TCP: Transmission Control Protocol

IP: Internet Protocol

HTTP: Hypertext Transfer Protocol

SMTP: Simple Mail Transfer Protocol

  1. Figures should be described and explained in detail. The elements of Figure 1 are not explained within the paper. You only write "see figure 1"; for example what Poseidon App is? It is not explained. The acronyms of Figure 2 should be explained (e.g., KAU, KCU, KOC), so that the reader can understand its meaning immediately. The sources of Figures 2-3 need a reference in the bibliography.

Reply->

Thank you for your valuable comments. I revised the paper as per your recommendation

Figure 1. Smart Renewable Energy and P2P (Peer-to-Peer) Blockchain Service

As shown in (see Figure 1), Retail allows users to buy and sell goods, energy, etc. using Carbon Impact. In addition, Back-End signed an electronic contract for Carbon Credit, which is a smart contract, which is generated by UserProfile, and can use trading rights for carbon transactions in all mobile apps called Eco-System.

Korea is certified by the government as a result of reducing greenhouse gas emissions outside the workplaces of companies subject to allocation through the Korean Offset Credit (KOC), and can be converted into offset emission rights. This meaning is called KOC. In addition, the Korean Allowance Unit (KAU) means the emission rights allocated by the government to the allocated company. And offset emission unit (KCU) refers to the emission rights converted by the allocated company for external business certification performance.

  1. The source of Table 1 needs a reference. Moreover, some parts of such table have a different font.

Reply->

Thank you for your valuable comments. I revised the paper as per your recommendation

I unified the font in Table 1. And I put in the source of the table.

(Source: Software Policy Institute, 2018)

  1. At the beginning of Section 2 you show a list of papers published about a certain topic. It is unnecessary, because the reader can retrieve this information from the bibliography. Moreover, you cite 4 papers, but the references reported at the end of the paragraph are only 3 and wrong "[3-5]". Please, check.

-> Reply

Thank you for your valuable comments. I revised the paper as per your recommendation.

The references were changed to [3-6], citing a total of four.

6.Laura Franke, Marco Schletz and Søren Salomo, “Designing a Blockchain Model for the Paris Agreement’s Carbon Market Mechanism”, Sustainability, 2020, Vol.12, 1068.

  1. The references should be written according to the order of appearance in the paper. Please, see [36-38] (line 140). Please, check the overall paper.

Reply->

Thank you for your valuable comments. I revised the paper as per your recommendation.

And I revised the reference.

  1. The subsections 2.2.1-2.2.6 need references, especially as regards the numerical data that are reported.

Reply->

Thank you for your valuable comments. I revised the paper as per your recommendation.

  1. UNGlobalCompact. UNSustainableDevelopmentGoals. 2016. Availableonline: https://www.unglobalcompact. org/what-is-gc/our-work/sustainable-development/sdgs/17-global-goals(accessedon15March2019

  1. The sentence "The types of governance are largely blockchain governance, project governance, and fund governance." needs to be better explained.

Reply->

Thank you for your valuable comments. I revised the paper as per your recommendation.

Blockchain governance also proceeds in the form of voting. Of course, voting can be used not only in voting but also in other ways, but also in terms of efficiency, voting is commonly used. The most representative blockchain determines the direction of the network through BIP, EIP, and developer forums, and various communities, and in addition, various blockchain manage the network in slightly different ways. The meaning of project governance is that blockchain networks are a world. Based on this world, we can share all-chain governance and off-chain governance. On-chain governance, the former, refers to a form in which governance can be achieved within a chain without any other elements other than blockchain network components, and blockchain networks can be considered complete as an independent world. Conversely, if governance is done outside the network and it has a direct impact on the blockchain network, this form of governance can be said to be off-chain governance.And fund governance is an act to create a fund to raise the resources needed to create a blockchain ecosystem.

  1. Figure 5 is well-built and interesting, but it should be better explained in the paper. Which is the flow? The mechanism? How do the various parts interact with each other?

Reply->

Thank you for your valuable comments. I revised the paper as per your recommendation.

Figure 5. Carbon Trading AI Blockchain Platform

The above (see Figure 5)stores MRV data, GHG data, and unstructured/structured data in the Web Application Server (WAS). The natural network is mapped to reflect this. And these analyzed engines are stored in the Deep Learning repertoire. These stored databases are stored in the RTE server, and the data analyzes and delivers databases that can analyze real-time data.

â‘  Transfers MRV data to Web Application Server (WAS)

â‘¡ GHG data to WAS

â‘¢ Send structured and unstructured data to WAS

â‘£ Transferring Natural Network Data

⑤ Application of Blockchain Agreement Algorithm

â‘¥ Save to Deep Learning data Repository

⑦ ExceptionSaving on a Processable Server

â‘§ Real time Enterprise (RTE)Save to server

⑨ Real-time DB, divided into storage DBs

â‘© Store on final set database server

⑪ Displayed in dashboard

  1. In line 524, you talk about Joel Mongro; a reference is needed.

Reply->

Thank you for your valuable comments. I revised the paper as per your recommendation

Reference [44] has been added

  1. Corea F. (2019) AI and Blockchain. In: An Introduction to Data. Studies in Big Data, vol 50. Springer, Cham

  1. The conclusions should only clarify and summarize the main aspects of the article. Therefore, I think that Figures 13-16 should be placed and explained before starting the conclusions.

Reply->

Thank you for your valuable comments. I revised the paper as per your recommendation

  1. Conclusion and Future Works is change Discussion and Future Works

Conclusion and Future Works -> Discussion and Future Works

And add 5. Conclusion

  1. Conclusion

While many topics such as consensus algorithm (PoW, PoS, etc.) and coin economy (how to distribute coin/talk to whom, etc.) and scalability are discussed in the blockchain-based decentralized network, governance related to how to make decisions is not likely to be discussed well. Thus, in this paper, the centralized network talks about why the UN-Gs Governance is important. In general, planning a product is a process of looking at what the user's potential demand in the market is, and then designing a solution that can meet this demand (i.e., address the user's problems). Even if you identify the hidden needs of the market, however, it is very difficult to create a product that can accurately address those needs, i.e., "shooting at what the market wants.” In most cases, no matter how well-prepared and released the product is, it does not capture the demand of the market fully but meets it only partially. Because it is so difficult to predict demand in the market, instead of planning elaborately at first with high cost and effort, the company makes a minimal functional product (MVP) that can meet the expected market demand even if quality drops a little, and then quickly modifies the product according to the market response to find a product-market combination at the beginning of the business. Such need to make products that satisfy what the market wants is the same for decentralized projects based on blockchain. It is equally difficult to satisfy the market, so no matter how sophisticated the blockchain characteristics and token Explosions are before the start, the market rarely flows in the direction predicted before the launch of the network; in most cases, the market will flow in a different direction from the originally predicted one. It would be the same for blockchain-based decentralized projects that have to make products satisfying the needs of the market, but it is not easy to satisfy the market; thus, no matter how sophisticated the blockchain characteristics and token Explosions are before the start, the market rarely flows in the direction predicted before the launch of the network; in most cases, the market runs in a different direction from the originally predicted one. And carbon credits are currently a very difficult technology to physically verify with emission rights. Therefore, this paper aims to establish a governance system for UN-SDGs to verify these carbon emissions. In order to establish this governance system, technology independence must be achieved first. Technology is something that needs to be reliable. As such, this paper designed the architecture to enhance the performance information of the trust-based blockchain. It is also equipped with an expanded version of the DPoS method, an agreement algorithm. Second, the verification encryption algorithm needed to build the engine of the blockchain is discussed. The verification amberization algorithm is a performance, trust-based encryption algorithm. And the sixth is the convergence of artificial intelligence and blockchain algorithms. It discusses the mechanism of using blockchain and artificial intelligence algorithms to make sure that carbon emission rights are the correct or wrong answer. This paper now says that carbon credits were previously converted into B2B markets from carbon emission exchanges, but are now expanding to P2P markets. Therefore, if this P2P market becomes a P2P market, it is a paper for establishing and verifying carbon emission block chain governance to verify carbon emission rights.

  1. Your contribution must be clearer. Are there articles in the literature that deal with the use of blockchain technology in the carbon trading? What is introduced differently in your paper? There are some papers in the literature about this topic. A brief comparison should improve the overall quality of your manuscript. Some examples are:

Pan et al. (2019), Application of Blockchain in Carbon Trading, Energy Procedia;

Khaqqi et al. (2018), Incorporating seller/buyer reputation-based system in blockchain-enabled emission trading application, Applied Energy.

These are just some examples, you don't necessarily have to mention these.

Reply->

Thank you for your valuable comments. I revised the paper as per your recommendation

3.5 Blockchain Validation Comparison

It is necessary to verify through a test net to develop dynamic (within 1 second of overhead) multi-channel blockchain technology that supports real-time transaction parallel processing through graph profiling of carbon emission blockchain event-based health management.

  1. Real-time Multi-Channel Communications Protocol
  2. Improve transaction parallelism using profile information and speculative execution
  3. Event-based multi-chain health management system development
  4. Hierarchical Block Chain Creation and Maintenance System
  5. Parallel processing and optimization of transactions using dependent graphs
  6. Develop dynamic channel allocation and merge technologies

Finally, carbon verification is required among carbon emission block chain verification systems based on End Point.

  1. Major Functions

- Multi-chain health management system

- Multi-chain block creation and maintenance system

- Multi-channel communication protocol module

- Virtual machines that process transactions in parallel through profiling and speculative execution

- Dynamic channel allocation and merge algorithm

- Blockchain test net with scalability solution for this task

- Mainnet (SW)

  1. Major skills

- Blockchain-based NoSQL real-time database and health path event function structure

- Transaction separation processing system based on channel-to-channel dependencies

- Blockchain virtual machines that can execute speculation through parallel processing and profiling of transactions in a channel

- A system that dynamically manages channels using profile information to ensure uniform transaction throughput for each channel.

  1. Comparison of key performance data

- Transaction processing speed 15,000 TPS

- 100 concurrent transaction types

- 30 seconds of throughput for transactions dependent on more than one channel

- 10x the in-channel transaction parallel throughput

- Overhead 2 seconds for dynamic channel management

  1. Is there experimental evidence about the effectiveness of dAPP? Please, deepen this aspect.

Reply->

Thank you for your valuable comments. I revised the paper as per your recommendation

Figure 13. Number of current transaction types performance data (average number 5)

Measures the time it takes for transactions to process that change the state managed by more than one channel for verification of carbon blockchain. It also compares the transaction throughput (TPS) measured in parallel with the transaction throughput (TPS) in the control group, which processes the transactions within you continuously. And (see Figure 13)shows the average value of 12,000 averages when these performance data are rotated five times.

Figure 14. Number of current transaction types performance data (average number 100)

In addition, the test bed network environment consisting of 10 nodes is established and the performance data is measured with approximately 15,000 TPS performance data values when measured as shown in (see Figure 14)about 100 times when the test bed network environment is constructed with the same 10 nodes as the client sending the transaction request to each channel through a non-numeric method. Thus, the results of measuring the number of channels that can be generated while maintaining the appropriate number of transactions in a testbed network environment consisting of these 50 identical nodes show that the performance information is stable as the average value is paid 100 times.

Reviewer 2 Report

I do not see much improvements from the previous version. Please fully answer to all comments provided at the first round. 

Author Response

(x) Moderate English changes required

Reply->

Thank you for your valuable comments.

At the same time, to cover the drawbacks, the contribution parts have been included in every possible section while correcting the contents with the help of a native English speaker to improve readability within a limited time frame. The revised or added parts are being highlighted in Red/Green for your possible re-review.

Does the introduction provide sufficient background and include all relevant references? Must be improved

( ) ( ) (x) ( )

Is the research design appropriate? Not applicable

( ) ( ) ( ) (x)

Are the methods adequately described? Not applicable

( ) ( ) ( ) (x)

Are the results clearly presented? Must be improved

( ) ( ) (x) ( )

Are the conclusions supported by the results? Not applicable

( ) ( ) ( ) (x)

Comments and Suggestions for Authors

I do not see much improvements from the previous version. Please fully answer to all comments provided at the first round.

Reply->

Thank you for your valuable comments. I revised the paper as per your recommendation. I will make the best of the comments I made in the first round.

I will make the best of the comments I made in the first round.

  1. Introduction

Also, we use blockchain to verify carbon emission rights in our efforts to manage energy. These efforts are taking a little further into the verification system for blockchain. But so far, I think the blockchain carbon emission system is in its early stages. Therefore, in order to prevent hacking and strengthen security vulnerabilities, the available blockchain network protocol will be applied and changed to the appropriate blockchain network protocol for the greenhouse gas comprehensive information center.

3.2. Research Methodology

Blockchain governance also proceeds in the form of voting. Of course, voting can be used not only in voting but also in other ways, but also in terms of efficiency, voting is commonly used. The most representative blockchain determines the direction of the network through BIP, EIP, and developer forums, and various communities, and in addition, various blockchain manage the network in slightly different ways. The meaning of project governance is that blockchain networks are a world. Based on this world, we can share all-chain governance and off-chain governance. On-chain governance, the former, refers to a form in which governance can be achieved within a chain without any other elements other than blockchain network components, and blockchain networks can be considered complete as an independent world. Conversely, if governance is done outside the network and it has a direct impact on the blockchain network, this form of governance can be said to be off-chain governance.And fund governance is an act to create a fund to raise the resources needed to create a blockchain ecosystem.

Describe the architecture for introducing blockchain for MRV application in the GHG reduction process. It provides a basic foundation for MRV according to the national CDM methodology. In addition, Most nations is developing and equipped with a greenhouse gas reduction zone. Also, there is a lot of data every year, including statistics on greenhouse gas emissions. However, this is because it is the basis for each country to manipulate MRVs. Thus, in order to reduce this GHG, the reduction target must be achieved. There is a lot of discussion about this reduction target. So far, however, the state has arbitrarily granted carbon credits. In addition, each country's guidelines exist but no clear standardization is made. Thus, the Climate Change and Energy Program emerged as a program to manage climate change and energy. But these programs are also being challenged to manipulate or by malicious hackers.

Therefore, this research was conducted with Blogain Agreement Algorithm and Artificial Intelligence Deep Learning Engine for research through verification of GHG reduction and MRV.

Figure 5. Carbon Trading AI Blockchain Platform

The above (see Figure 5)stores MRV data, GHG data, and unstructured/structured data in the Web Application Server (WAS). The natural network is mapped to reflect this. And these analyzed engines are stored in the Deep Learning repertoire. These stored databases are stored in the RTE server, and the data analyzes and delivers databases that can analyze real-time data.

â‘  Transfers MRV data to Web Application Server (WAS)

â‘¡ GHG data to WAS

â‘¢ Send structured and unstructured data to WAS

â‘£ Transferring Natural Network Data

⑤ Application of Blockchain Agreement Algorithm

â‘¥ Save to Deep Learning data Repository

⑦ ExceptionSaving on a Processable Server

â‘§ Real time Enterprise (RTE)Save to server

⑨ Real-time DB, divided into storage DBs

â‘© Store on final set database server

⑪ Displayed in dashboard

Reviewer 3 Report

Some of my previous main remarks were not taken into account. Namely:

The relevance of the paper in the introduction is described very briefly,
amost no evidence of other similar researches and their results is
provided.

In the chapter "Background Knowledge" under the subsection 2.2
"UN Sustainable Development Goals" following 6 short subsections
are provided. I recommend not to split that subsection into so smaller
subsections. Eventhoug the Sustainable Develompent Goals are
imporant for this research, it should be more focused on the essence of
this paper, not describint the problems of the clean water and sanitation,
mitigation of inequality.

The research methodology is not described clearly: what are the steps
and their sequence for the research? There is no clear point in the paper
where the survey description starts.

Author Response

Comments and Suggestions for Authors

Some of my previous main remarks were not taken into account. Namely:

The relevance of the paper in the introduction is described very briefly,

amost no evidence of other similar researches and their results is

provided.

In the chapter "Background Knowledge" under the subsection 2.2

"UN Sustainable Development Goals" following 6 short subsections

are provided. I recommend not to split that subsection into so smaller

subsections. Eventhoug the Sustainable Develompent Goals are

imporant for this research, it should be more focused on the essence of this paper, not describint the problems of the clean water and sanitation, mitigation of inequality.

Reply->

Thank you for your valuable comments. I revised the paper as per your recommendation. I will make the best of the comments I made in the first round.

2.2.1. Water and Sanitation

Clean and available water is essential for a healthy ecosystem and for human health. There is also enough fresh water on Earth for this. Due to economic difficulties and poor infrastructure, however, millions of people, including children, die each year from water shortages and sanitation problems. Lack of water, worsening water quality, and inadequate sanitation have a negative impact on food security, livelihood, and educational opportunities for poor families around the world. Today, more than two billion people live at risk of desalination; by 2050, at least one in four will be affected by chronic, repetitive water shortages. In particular, drought affects the poorest countries, exacerbating hunger and malnutrition. Fortunately, significant progress has been made in drinking water and sanitation over the past decade, with more than 90 percent of the world's population now having access to improved drinking water supply. In order to improve water quality and improve access to supply, many developing countries in sub-Saharan Africa, Central Asia, South Asia, East Asia, and Southeast Asia need to expand their investment in the management of water systems and sanitation in their regions.

2.2.3. Industrialization, Innovation, and Infrastructure

Investment in infrastructure, such as transportation, irrigation, energy, information, and communications technologies, is critical to achieving sustainable development and empowering the community in many countries. It has long been recognized that productivity, income growth, and improved health and education require investment in infrastructure. Manufacturing is a major driver of economic growth, employment, and social stability. North America and Europe have added $4500 in manufacturing value per person, whereas the least developed countries have only about $100. Another important factor to consider is the carbon dioxide emissions in the manufacturing process. Over the past decade, carbon emissions have decreased in many countries, but the decline has not taken place globally. Advancing technology should involve efforts to achieve environmental goals such as increasing resources and energy efficiency. Without technology and innovation, industrialization is not going to happen; without industrialization, development is not going to happen. More investment is needed in cutting-edge products that impact manufacturing for greater efficiency, including focusing on mobile services that enhance human-to-human connectivity.

2.2.4. Mitigation of Inequality

The international community has made significant progress in fighting poverty. Efforts are underway to reduce inequality in the poorest, inland, and developing countries and small islands. Nonetheless, there is still inequality in health, education, and asset management. There is a growing consensus that economic growth, if not comprehensive and not inclusive of the three elements of sustainable development at the economic, social, and environmental levels, is not enough to reduce poverty. Fortunately, income inequality has been reduced both within the country's borders and within the country. At least 60 out of 94 countries that have data have seen their per capita income grow faster than the national average, and some progress has been made in creating favorable conditions for exports from underdeveloped countries. To reduce inequality, policies should be universal, in principle, and attention should be paid to the needs of vulnerable and underprivileged people. In addition to increasing voter turnout in developing countries within the IMF, we need to continue to be tax-free and export-driven in developing countries. Finally, innovation can help reduce the high cost of money transfers for migrant workers.

2.2.5. Sustainable Cities and Communities

Cities served as hubs for ideas, commerce, culture, science, productivity, and social development, enabling people to develop socially and economically. As the number of people living in cities is expected to grow to 5 billion by 2030, it is important to have efficient urban planning and management approach to address the challenges posed by urbanization. There are many challenges in maintaining the city in a way that creates jobs and mutual prosperity without putting pressure on land and resources. Common urban problems include low management operating costs, housing shortage, reduced living infrastructure, and increased air pollution in addition to rapid urbanization, which gives rise to problems such as safe collection and disposal of solid waste in cities but can be overcome in such a way that development and growth can continue by improving resource use and reducing pollution and poverty. One such example is the increase in urban garbage collection. Cities of the future also need to provide access to basic services, energy, housing, transportation, etc.

2.2.6. Responsible Consumption and Production

Sustainable consumption and production boost resources and energy efficiency, create a sustainable infrastructure, provide basic services as well as eco-friendly and appropriate jobs, and ensure a better quality of life. Practicing sustainable consumption and production will help achieve the overall development plan, reduce future economic, environmental, and social costs, enhance industrial competitiveness, and reduce poverty. Currently, metal consumption is on the rise and is the largest in East Asia. Countries around the world also continue to address issues related to air, water, and soil pollution. Sustainable consumption and production are aimed at "getting more and doing better for less," so that the welfare benefits from economic activities can be increased by reducing waste and pollution while improving the quality of life. The focus should also be on running the supply chain, involving everyone from producers to end consumers. These include educating consumers on sustainable consumption and lifestyle, providing them with appropriate information through standards and labels, and engaging in sustainable public procurement.

is delete.

But add is

2.2.2. Response to Climate Change

Climate change is now affecting all countries across the continent. It is disrupting the national economy, affecting life, and causing great losses to humanity, communities and countries today and in the future. Weather patterns are changing, sea levels are rising, climate change is frequent, and greenhouse gas emissions are now at their highest levels. If no action is taken, sea level temperatures are expected to rise above an average of 3 degrees this century. This has the most effect on the poor and vulnerable. By seeking economic and scalable solutions, the nation can become a cleaner and more resilient economy. The pace of change is accelerating as more people seek ways to use renewable energy and reduce and adapt to greenhouse gas emissions. But climate change is a transnational global challenge. We need an international level of improvement to help developing countries move toward a low-carbon economy. To strengthen international response to the threat of climate change, countries adopted the Paris Convention at the Conference of Parties to the United Nations Framework Convention on Climate Change (COP21), which took effect in November 2016. Under the agreement, all countries agreed to limit the Earth's temperature rise to less than 2 degrees Celsius. In April 2018, 175 countries ratified the Paris Agreement and 10 developing countries submitted their first national response plans to combat climate change.

The research methodology is not described clearly: what are the steps

and their sequence for the research? There is no clear point in the paper

where the survey description starts.

Reply->

Thank you for your valuable comments. I revised the paper as per your recommendation.

3.2. Research Methodology

Blockchain governance also proceeds in the form of voting. Of course, voting can be used not only in voting but also in other ways, but also in terms of efficiency, voting is commonly used. The most representative blockchain determines the direction of the network through BIP, EIP, and developer forums, and various communities, and in addition, various blockchain manage the network in slightly different ways. The meaning of project governance is that blockchain networks are a world. Based on this world, we can share all-chain governance and off-chain governance. On-chain governance, the former, refers to a form in which governance can be achieved within a chain without any other elements other than blockchain network components, and blockchain networks can be considered complete as an independent world. Conversely, if governance is done outside the network and it has a direct impact on the blockchain network, this form of governance can be said to be off-chain governance.And fund governance is an act to create a fund to raise the resources needed to create a blockchain ecosystem.

Describe the architecture for introducing blockchain for MRV application in the GHG reduction process. It provides a basic foundation for MRV according to the national CDM methodology. In addition, Most nations is developing and equipped with a greenhouse gas reduction zone. Also, there is a lot of data every year, including statistics on greenhouse gas emissions. However, this is because it is the basis for each country to manipulate MRVs. Thus, in order to reduce this GHG, the reduction target must be achieved. There is a lot of discussion about this reduction target. So far, however, the state has arbitrarily granted carbon credits. In addition, each country's guidelines exist but no clear standardization is made. Thus, the Climate Change and Energy Program emerged as a program to manage climate change and energy. But these programs are also being challenged to manipulate or by malicious hackers.

Therefore, this research was conducted with Blogain Agreement Algorithm and Artificial Intelligence Deep Learning Engine for research through verification of GHG reduction and MRV.

3.5. Blockchain Validation Comparison

It is necessary to verify through a test net to develop dynamic (within 1 second of overhead) multi-channel blockchain technology that supports real-time transaction parallel processing through graph profiling of carbon emission blockchain event-based health management.

  1. Real-time Multi-Channel Communications Protocol
  2. Improve transaction parallelism using profile information and speculative execution
  3. Event-based multi-chain health management system development
  4. Hierarchical Block Chain Creation and Maintenance System
  5. Parallel processing and optimization of transactions using dependent graphs
  6. Develop dynamic channel allocation and merge technologies

Finally, carbon verification is required among carbon emission block chain verification systems based on End Point.

  1. Major Functions

- Multi-chain health management system

- Multi-chain block creation and maintenance system

- Multi-channel communication protocol module

- Virtual machines that process transactions in parallel through profiling and speculative execution

- Dynamic channel allocation and merge algorithm

- Blockchain test net with scalability solution for this task

- Mainnet (SW)

  1. Major skills

- Blockchain-based NoSQL real-time database and health path event function structure

- Transaction separation processing system based on channel-to-channel dependencies

- Blockchain virtual machines that can execute speculation through parallel processing and profiling of transactions in a channel

- A system that dynamically manages channels using profile information to ensure uniform transaction throughput for each channel.

  1. Comparison of key performance data

- Transaction processing speed 15,000 TPS

- 100 concurrent transaction types

- 30 seconds of throughput for transactions dependent on more than one channel

- 10x the in-channel transaction parallel throughput

- Overhead 2 seconds for dynamic channel management

Figure 13. Number of current transaction types performance data (average number 5)

Measures the time it takes for transactions to process that change the state managed by more than one channel for verification of carbon blockchain. It also compares the transaction throughput (TPS) measured in parallel with the transaction throughput (TPS) in the control group, which processes the transactions within you continuously. And (see Figure 13)shows the average value of 12,000 averages when these performance data are rotated five times.

Figure 14. Number of current transaction types performance data (average number 100)

In addition, the test bed network environment consisting of 10 nodes is established and the performance data is measured with approximately 15,000 TPS performance data values when measured as shown in (see Figure 14)about 100 times when the test bed network environment is constructed with the same 10 nodes as the client sending the transaction request to each channel through a non-numeric method. Thus, the results of measuring the number of channels that can be generated while maintaining the appropriate number of transactions in a testbed network environment consisting of these 50 identical nodes show that the performance information is stable as the average value is paid 100 times.

Round 2

Reviewer 1 Report

The paper was slightly improved, but the authors only partially responded to my requests.

1. First of all, the paper needs further control of the English language. I recommend consulting an English native speaker.

2. At the end of Section 1, an introductory part to the rest of the paper is missing. You should use the classic formula: "The remainder of this paper is organized as follows ...". In this way, the reader has the possibility of realizing what he will read on the following pages.

3. At the beginning of Section 2, you cite some articles, but there are some mistakes. You mention "Luo et al." which is the reference n. [8], but at the end of the paragraph you write [3-6], excluding [8]

4. Figure 2 is not adequately commented.

5. In subsection 2.2.1 there are no references to the data, which I had requested instead. In addition, several subsections have been removed (e.g., water and sanitation, industrialization, innovation and infrastructure) but I can not understand the reason.

6. Table 1, Figures 2,3 need a reference in the bibliography.

7. The references should be ordered according to the appearance within the paper. Please see line 154 "[36-38].

8. A comparison with the other papers in the literature, which address a similar topic, would make your contribution clearer, but it is missing.

Many of the above mentioned requests relate to the previous review process, but have not been met.

Author Response

Comments and Suggestions for Authors

The paper was slightly improved, but the authors only partially responded to my requests.

  1. First of all, the paper needs further control of the English language. I recommend consulting an English native speaker.

Reply->

Thank you for your valuable comments. I revised the paper as per your recommendation. I also respect your opinion. I will write it with care for English expression.

The revised or added parts are being highlighted in red for your possible re-review.

  1. At the end of Section 1, an introductory part to the rest of the paper is missing. You should use the classic formula: "The remainder of this paper is organized as follows ...". In this way, the reader has the possibility of realizing what he will read on the following pages.

Reply->

Thank you for your valuable comments. I revised the paper as per your recommendation. I also respect your opinion. I will write it with care for English expression.

Add)

  1. Introduction

In addition, this study provides a scope for the purpose of verifying carbon emission rights using blockchain to conduct transactions between individuals and for the performance of trading carbon emission rights. And the method aims to instantiate blockchain performance data, Transaction per Second (TPS), and to ensure the transparency and integrity of carbon credits for future P2P transactions.

  1. At the beginning of Section 2, you cite some articles, but there are some mistakes. You mention "Luo et al." which is the reference n. [8], but at the end of the paragraph you write [3-6], excluding [8]

Reply->

Thank you for your valuable comments. I revised the paper as per your recommendation. I also respect your opinion.

Add)

2.1. Carbon Emission

[6] is Delete

  1. Figure 2 is not adequately commented.

Reply->

Thank you for your valuable comments. I revised the paper as per your recommendation. I also respect your opinion. I'll explain it to the picture.

Add)

In addition, the market for carbon credits in Revenue, 8.4 percent in 2015, has grown about three times to about 22 in 2018. And KOC trading volume is 1,296 in 2018, compared with just three years ago. And the total volume of KAU trading also stood at 21,242, with carbon credits showing transactions between many companies.

  1. In subsection 2.2.1 there are no references to the data, which I had requested instead. In addition, several subsections have been removed (e.g., water and sanitation, industrialization, innovation and infrastructure) but I can not understand the reason.

Reply->

Thank you for your valuable comments. I revised the paper as per your recommendation. I also respect your opinion. This paper deals with the carbon emission rights of UN-SDGs, so it has been removed but I will add it again.

Add)

2.2.3. Poverty eradication

The global poverty rate has been halved since 2000, but one in 10 developing countries still lives below the $1.9-a-day international poverty line, and there are millions of people making less than a day's living expenses. Despite outstanding progress in East Asia and Southeast Asia, 42 percent of sub-Saharan Africa's population is still suffering from extreme poverty. Poverty means more than just a lack of income and resources to ensure a sustainable livelihood. These include hunger and malnutrition, restrictions on education and living services, social discrimination and exclusion, and restrictions on decision-making participation. Economic development must have a comprehensive goal to provide sustainable jobs and improve the inequality structure. Social protection needs support to alleviate the suffering of countries at risk of disaster and to overcome economic crises, and these systems will help strengthen the responsiveness of those suffering from unexpected cost losses in the event of disaster and, ultimately, end extreme poverty in areas of absolute poverty.

2.2.4. Famine species

It's time to rethink how we grow, distribute and consume food. If done properly, agriculture, forestry and fishing can provide everyone with nutritious food, generate a significant level of income, and at the same time support people-centered rural development and protect the environment. Now biodiversity is rapidly decreasing due to damage to soil, fresh water, sea, and forests. Climate change is having a devastating effect on the resources we depend on, and it's increasing the risk of disasters such as drought and flooding. Many farmers are no longer able to make ends meet on their land and have to move to cities to find opportunities, and poor food security is seriously undernourished, hurting the growth of millions of children or shortening their lifespan. The world's undernourished population is expected to be 815 million, and an additional 2 billion by 2050, which requires fundamental changes in the world's food and agriculture systems. Investment in agriculture is essential to the development of agricultural productivity, and sustainable food production systems are necessary to reduce the risk of malnutrition.

2.2.5. Health and well-being

Ensuring a healthy life for all ages and promoting welfare are essential to sustainable development. While significant progress has been made, including increasing human life expectancy and decreasing infant and pregnant women's death rates, a professional improvement in the delivery system is needed to reduce the death rate of fewer than 70 children per 100,000 people by 2030. In order to achieve the goal of reducing early death rates from non-inflammatory diseases by one-third by 2030, efficient techniques are encouraged to use clean oils in cooking, and training on the dangers of cigarettes is needed. It also takes a lot of effort to eradicate a variety of diseases and to address the ongoing health problems. By focusing on providing effective funding for the healthcare system, improving sanitation, increasing access to medical services, and providing information on preventing environmental pollution, we can make great strides in saving millions of lives.

2.2.6. Quality education

Quality education is the foundation for sustainable development. In addition to improving the quality of life, providing comprehensive education also helps local people have the creative thinking they need to find innovative solutions to international issues. Currently, more than 265 million children have dropped out of school and 22 percent of them are only elementary school students. In addition, even children who go to school lack basic skills such as reading and calculating Over the past decade, efforts have been made to increase access to full-course education and to increase women's school enrollment. The eradication of basic illiteracy has increased dramatically, but more effort is needed to achieve universal educational goals and progress. For example, while the equality of primary education between men and women has been achieved, few countries have achieved that goal in the whole course of education. The reason for the lack of quality education is linked to capital issues, such as the lack of trained teachers, poor school facilities, and relatively poor opportunities for rural children. In order to provide quality education to children from poor families, we need to invest in educational scholarships, teacher training workshops, school building and water quality improvement, and school electrical facilities.

2.2.7. Gender Equality

While the Millennium Development Goals (including achieving universal primary education, MDGs) have made progress in gender equality and women's rights and interests, women and girls continue to suffer discrimination and violence throughout the world. Gender equality is not only a basic human right, but a necessary foundation for a sustainable world that pursues peace and prosperity. One in five women between the ages of 15 and 49 stated that they had been victimized by an intimate partner within 12 months of the survey date, and 49 countries currently do not have laws to protect women from domestic violence. Over the past decade, child marriages have been on the decline, and progress has been made, with the harmful practice of female genital resection (female circumcision) decreasing by 30%, but more intensive efforts are needed to eradicate this practice. Providing women and girls with education, health care, quality jobs, and increasing their participation in the political and economic decision-making process will bring overall benefits to the sustainable economy, society and humanity. Therefore, establishing a new legal system for equality of women in the workplace and eradicating harmful practices against women is critical to ending gender discrimination in many countries around the world.

  1. Table 1, Figures 2,3 need a reference in the bibliography.

Reply->

Thank you for your valuable comments. I revised the paper as per your recommendation. I also respect your opinion. I added a reference to Figure 2, 3 and Table 1

Add)

  1. Korean Gorverment,"2nd National General Management Plan for Greenhouse Gas Statistics (2020-2024)" Magagine, 30-34, 2020.
  2. Dohyun Lee, Seulki Han and Jiyong Kim, "Economic and Environmental Assessment of a Renewable Stand-Alone Energy Supply System Using Multi-objective Optimization", 55(3), 332-340, 2017.
  3. Jun-Ho Huh, Seong-Kyu Kim, "The blockchain consensus algorithm for viable management of new and renewable energies", 11-11, 3184, 2019.

  1. The references should be ordered according to the appearance within the paper. Please see line 154 "[36-38].

Reply->

Thank you for your valuable comments. I revised the paper as per your recommendation. I also respect your opinion. I wrote [36-38] incorrectly. It was removed.

  1. A comparison with the other papers in the literature, which address a similar topic, would make your contribution clearer, but it is missing.

Many of the above mentioned requests relate to the previous review process, but have not been met.

Reply->

Thank you for your valuable comments. I revised the paper as per your recommendation. I updated everything.

Reviewer 2 Report

This is not qualified for an academic paper. 

Also I do not see any fundamental improvements from the author's previous version. When writing a reply to reviewer's comments, authors need to answer each comments in principle. 

Author Response

Comments and Suggestions for Authors

This is not qualified for an academic paper.

Also I do not see any fundamental improvements from the author's previous version. When writing a reply to reviewer's comments, authors need to answer each comments in principle.

Reply->

Thank you for your valuable comments. I revised the paper as per your recommendation. I also respect your opinion. I will update the previous comment as much as possible.

The revised or added parts are being highlighted in red for your possible re-review.

ADD)

3.2 . Research Methodology

In addition, the types of governance structures are largely divided into blockchain governance, project governance, and fund governance.

1) Blockchain Governance Structure

It refers to an algorithm that verifies accurate data and various data using blockchain. This level of governance is the stage in which all blockchain governance systems are established. It is also in the process of utilizing existing verification algorithms.

2) Structural Structure Project

The project governance structure is used to establish the ICT and environmental governance systems required to build and implement carbon emission reduction projects. The project for carbon reduction is a very important step and is related to blockchain agreement algorithms.

3) Fund Governance Structure

Fund governance refers to the blockchain fund governance structure that is needed when the financial sector funds are made when the carbon emission exchange is formed in future P2P transactions in the financial market of carbon emission rights.

3.3 Blockchain Process for Reducing GHG and MRV

emission rights in various and flexible ways in addition to trading emission rights when submitting emission rights in accordance with their target monthly greenhouse gas emission reduction activities. The purpose of flexible sex is to secure liquidity in the emission market by diversifying greenhouse gas reduction methods and inducing more greenhouse gas reduction activities by ensuring flexibility in the method of submitting emission rights. Types of flexible meker nism include banking and borrowing, recognition of early reduction performance, and certification of external reduction projects such as offset. In addition, to reduce this MRV, a verification platform that is interlinked with these policy segments must be a mix of goods. This is the same for the whole world. This paper used artificial intelligence blockchain for the verification platform in line with the process of GHG reduction and MRV reduction, and is expected to show a lot of trading volume if the carbon emission trading system comes out as an individual trading system in the future. Therefore, this paper is equipped with a process to perform these verification procedures. The process required for it is shown in (see Figure 4).

Figure 4. Verification Blockchain for MRV, GHG Data

1) Step 1

Large amounts of carbon credits will be created, and individuals will be able to purchase them in the future. Thus, it is the verification of the data required to reduce the MRV and GHG required.

2) Step 2

Blockchain is a step to propose a blockchain architecture that can achieve a very fast agreement according to Hyper PoR consensus algorithm and verify it by these consensus algorithms.

3) Step 3

The verification algorithm of the blockchain is used to verify each node and to verify it through work in the mesh network.

4) Step 4.

Obtain algorithms to agree and verify carbon emission rights. Carbon emission rights are the steps to verify carbon credits mined by individuals and exchanges necessary for this reduction.

5) Step 5

Store the final validated data divided into agreed nodes after the data consistency verification process. These storage steps are stored thoroughly prepared for hacking.

  1. Experimental Result

4.1 Experimental Environment

Establish an environment to obtain blockchain performance data.

system environment

- Server : HP ML-500- 6

- Network: 5G Bps

- Environment: AWS Clode

- Node: 100

- Database : IPFS

4.2. Experimental Condition

4.3. Blockchain Validation Comparison

Reviewer 3 Report

Thanks for corrections. Good luck.

Author Response

Comments and Suggestions for Authors

Thanks for corrections. Good luck.

Reply->

Thank you for your valuable comments. I revised the paper as per your recommendation. I also respect your opinion. I will update the previous comment as much as possible.

The revised or added parts are being highlighted in red for your possible re-review.

Round 3

Reviewer 1 Report

The authors responded only to some of my requests. Then, some points remain to be clarified:

1. At the end of Section 1, you need to introduce the content of the remainder of the paper. Please, write what your article is about, section by section.

2. At the beginning of Section 2, some references continue to be disregarded. For example, "Andoni et al." is indicated as reference [3], but in the bibliography it is erroneously marked as fourth. Please, check this part.

3. Figure 2 is not well described. The sentence "And KOC trading volume is 1,296 in 2018, compared with just three years ago" is not clear. 

4. The subsections 2.2.1 - 2.2.7 need references because there are some numerical data to be justified.

5. The subsection "3.5.7" should be "3.5.3", please check.

6. Please, insert a reference in the source of Table 1. 

Author Response

The authors responded only to some of my requests. Then, some points remain to be clarified:

  1. At the end of Section 1, you need to introduce the content of the remainder of the paper. Please, write what your article is about, section by section.

Reply->

Thank you for your valuable comments. I revised the paper as per your recommendation. I also respect your opinion. I updated it by section. The revised or added parts are being highlighted in red for your possible re-review.

ADD)

Chapter 1 discusses the problems of carbon emissions at UN-SDGs and artificial intelligence blockchain to verify the trading rights when future carbon emissions are traded between individuals. And when we introduce these blockchain, we talk about important consensus algorithm. Chapter 2 presents directions on how to apply blockchain in UN-SDGs in Realized Work, and specifically discusses IT application and blockchain application for carbon credits and renewable energy. Chapter 3 describes the UN SDGs' Performance and Blockchain Algorithms for Design and Implementation. While the size of the carbon emission trading system has grown to a low point, there is a lack of technology to limit it, so research has begun. So we are talking about the Blockchain Process for ReducingGHG and MRV methodology and the verification process in total of five stages. About Protocol Design for Hybrid Governance, blockchain talks about how to use and verify IT governance. Procedure for verifying and agreeing on Carbon Tracking AI Blockchain Platform, and methodology for applying Hybrid Governance Protocol Applicable to UN SDGs is presented. The procedure of Certificate-based Hybrid Blockchain and Authentication of Hybrid Blockchain based on Cryptographic Protocol are applied by applying this methodology. Certification of Hybrid Blockchain based on MAC Address, Authentication of Hybrid Blockchain based on ID/password, Hybrid Governance Protocol for Design Pattern, Hybrid Governance Protocol for Visitor Pattern, Hybrid Governance Protocol for Governance Protocol for Protocols, and Hybrid Governance Protocol for Protocols. It will then unveil the source code for the Implementation of the Hybrid Governance Protocol. Chapter 4 shows configuration for performance comparison.The actual performance comparison shows 15,000 TPS. Chapter 5 discusses the direction of future research and current problems. Chapter 6 concludes with a conclusion on the study of the application of blockchain carbon credits

  1. At the beginning of Section 2, some references continue to be disregarded. For example, "Andoni et al." is indicated as reference [3], but in the bibliography it is erroneously marked as fourth. Please, check this part.

Reply->

Thank you for your valuable comments. I revised the paper as per your recommendation. I also respect your opinion. I updated it by section.

ADD)

  1. Introduction's [3] has been added. And 2. It was modified to [4-6] in Background Knowledge.

  1. Figure 2 is not well described. The sentence "And KOC trading volume is 1,296 in 2018, compared with just three years ago" is not clear.

Reply->

Thank you for your valuable comments. I revised the paper as per your recommendation. I also respect your opinion. I updated the description.

ADD)

In addition, the total volume of emissions traded in both in-house and over-the-counter transactions during the branch line result period (2015.1-2018.8) was 86.2 million tons, with 37.5 million tons and 48.7 million tons traded in disability and over-the-counter transactions accounting for 44% and 56%, respectively. By emission rights, 66.6 million tons of KAU, 3.4 million tons of KCU and 16.2 million tons of KOC were traded, representing 77 percent, 4 percent and 19 percent of the total. The total volume of transactions per year increased by 208%, 246% and 134% year-on-year to 5.7 million tons in 2015, 11.9 million tons in 2016, 29.3 million tons in 2017, and 39.2 million tons in 2018, and the last year was similar in 2018, reflecting only the performance of transactions in the first half of the year. In addition, during the same period, the average price increased by 156%, 122%, and 106% year-on-year to 17,1799 won in 2016, 20,897 won in 2017, and 22,127 KRW in 2018, ending with a double increase compared to the initial average price in 2015. The average transaction price for the entire trading period was 20,279 KRW. Also, by emission rights, KAU was traded at 15,767 KRW, KOC was 16,703 KRW, and KAU was traded at a relatively higher price than other emission rights, and the market price was slightly higher than the over-the-counter price. Continued increases in transaction prices and expansion of trading volume also affected transaction payments, rising 324%, 333% and 142% on-year to 204.4 billion won in 2016, 612.3 billion won in 2017, and 868 billion KRW in 2018, from 63.1 trillion KRW in 2015, and combined, the total transaction amount was 1.7477 trillion KRW. KAU accounted for 81%, 3%, and 15%, respectively, with 1.4231 trillion won, KCU 54 billion won, and KOC 270.6 billion KRW, while the total transaction amount in the market and over-the-counter trading market was 781 billion KRW and 966.7 billion KRW, respectively, representing 45% and 55% of the share, similar to the share of the trading volume[8].

  1. The subsections 2.2.1 - 2.2.7 need references because there are some numerical data to be justified.

Reply->

Thank you for your valuable comments. I revised the paper as per your recommendation. I also respect your opinion. I added a reference.

ADD)

Ministry of Enviroment, "http://ncsd.go.kr/", UN Sustainable Development Goals web page, 2020.

  1. The subsection "3.5.7" should be "3.5.3", please check.

Reply->

Thank you for your valuable comments. I revised the paper as per your recommendation. I also respect your opinion. I changed section 3.5.7 to 3.5.3.

ADD)

3.5.3. Implementation of the Hybrid Governance Protocol

  1. Please, insert a reference in the source of Table 1.

Reply->

Thank you for your valuable comments. I revised the paper as per your recommendation. I also respect your opinion. I added a reference.

ADD)

  1. Jun-Ho Huh, Seong-Kyu Kim, "The blockchain consensus algorithm for viable management of new and renewable energies", Sustainability, 11(11), pp. 1-26,2019.

This manuscript is a resubmission of an earlier submission. The following is a list of the peer review reports and author responses from that submission.

Round 1

Reviewer 1 Report

This paper aims to make transaction platform with by applying blockchain technology to measure (?) carbon emission rights. Some researchers tried to bridge the idea of blockchain with climate change issues (including myself) but have not been successful so far in that profitable business model could not be applicable in designing an incentive system in every aspects of carbon markets (allocation, MRV, transaction, etc.)

This paper has too many problems in its contents so that I wonder if it can be improved by revisions. I hereby listed only some problematic issues not acceptable in any journals (if they claims to be academic journals)

As a person who is somewhat familiar to Blockchain technologies, I can see that the code in this article only lists a very common pattern on a typical blockchain that has little to do with carbon products. The author lists the code in the middle of the paper, which is very inappropriate and should be moved to the appendix so that the reader can understand it in the text.

“In fact, we can protect against carbon 776 emissions anomalies by using big data and artificial intelligence in mobile cloud environments.”

The authors listed only their arguments without any clear evidences, and aren't academics papers supposed to explain in detail how they can be proven?

As stated in the introduction, the MRV (Measuring / Verification process) of carbon reduction must be presumed to estimate/measure reliable carbon credits, which is not provided at all in this paper: if so, what is the contribution of this paper besides treating carbon credit as one of hundreds of Altcoins?

Blockchain applications in carbon trading should be the key to the application of MRV in the GHG reduction process. However, this paper only lists the general blockchain trading process, as can be seen in the presented reference, and does not provide any information on the achievement of the UN goals of carbon emissions and reduction.

The authors only take screenshots of specific apps to inflate expectations of blockchain applicability.

It does not meet the formal requirements of general academic papers, such as inserting figures in conclusion.

The level of English is very low even for non-natives.

Reviewer 2 Report

The relevance of the paper in the introduction is described very briefly, amost  no evidence of other similar researches and their results is provided. 

In the chapter "Background Knowledge" under the subsection 2.2  "UN Sustainable Development Goals" following 6 short subsections are provided. I recommend not to split that subsection into so smaller subsections. Eventhoug the Sustainable Develompent Goals are imporant for this research, it should be more focused on the essence of this paper, not describint the problems of the clean water and sanitation, mitigation of inequality.

There are two chapters called "Carbon emission": 2.1. (row 92) and 2.4. (row 276).  They have to be merged or one of them should be renamed. 

Row 263: the sentence starts with the lower case "enables implementation with actual code..." and it seems that the sentence is not full. 

Row 282: the same nmuber 32.5 billions is provided stating that the global emissions rose to 32.5 tons in 2017 from 32.5 billion tons. There is no logic here.

Row 360: not understandable sentence: "What's good for users can be bad for minors, and what's good for developers."

Rows 461-470 and 470-476 provide the same text.

Row 769: "carbon emission" has other font as the whole text.

The research methodology is not described clearly: what are the steps and their sequence for the research? There is no clear point in the paper where the survey description starts. 

Reviewer 3 Report

The topic of the manuscript is actual and it worth to be investigated. The evolution of the e blockchain technology has determined important changes in various domains. Applying blockchain in analyzing the carbon trading for UN sustainable development goal fulfillments represents a further step in understanding the topic. Despite its soundness, the manuscript needs major revisions:

First of all the authors must insert sources for every and each figure, if its own design or imported from other sources. Also, please insert source for Table 1

A literature review section is missing. The authors must present their finding opposite to the literature findings in the field. The manuscript is too descriptive for a research piece. The authors must be more argumentative for their work.

Results and discussion section could be merged.

Please considered inserting a new section of Research limits before Conclusions.